# Outer radiation belt and inner magnetospheric response to sheath regions of coronal mass ejections: A statistical analysis

Milla M. H. Kalliokoski[1], Emilia K. J. Kilpua[1], Adnane Osmane[1], Drew L. Turner[2], Allison N. Jaynes[3], Lucile Turc[1], Harriet George[1], and Minna Palmroth[1,4]

[1]Department of Physics, University of Helsinki, Helsinki, Finland
[2]Space Sciences Department, The Aerospace Corporation, El Segundo, California, USA
[3]Department of Physics and Astronomy, University of Iowa, Iowa City, Iowa, USA
[4]Finnish Meteorological Institute, Helsinki, Finland

**Correspondence:** Milla M. H. Kalliokoski (milla.kalliokoski@helsinki.fi)

**Abstract.** The energetic electron content in the Van Allen radiation belts surrounding the Earth can vary dramatically on several timescales, and these strong electron fluxes present a hazard for spacecraft traversing the belts. The belt response to solar wind driving is yet largely unpredictable and especially the direct response to specific large-scale heliospheric structures has not been considered previously. We investigate the immediate response of electron fluxes in the outer belt to driving by sheath regions preceding interplanetary coronal mass ejections and the associated wave activity in the inner magnetosphere. We consider events from 2012 to 2018 in the Van Allen Probes era to employ the energy and radial distance resolved electron flux observations of the twin spacecraft mission. We perform a statistical study of the events using superposed epoch analysis, where the sheaths are superposed separately from the ejecta and resampled to the same average duration. Our results show that wave power of ultra-low frequency Pc5 and electromagnetic ion cyclotron waves, as measured by a geostationary GOES satellite, is higher during the sheath than during the ejecta. However, the level of chorus wave power, measured by Van Allen Probes, remains approximately the same due to similar substorm activity during the sheath and ejecta. Electron flux enhancements are common at low energies ($< 1$ MeV) throughout the outer belt ($L = 3$–$6$), whereas depletion occurs predominantly at high energies for high radial distances ($L > 4$). Distinctively, depletion extends to lower energies at larger distances. We suggest that this $L$-shell and energy dependent depletion results from magnetopause shadowing dominating the losses at large distances, while wave-particle interactions dominate closer to the Earth. We also show that non-geoeffective sheaths cause significant changes in the outer belt electron fluxes.

## 1   Introduction

The Van Allen radiation belts are toroidal regions surrounding the Earth that trap charged particles in the geomagnetic field (e.g., Van Allen, 1959). Traditionally, the belts are divided into two zones of energetic populations: the relatively stable inner belt that is dominated by high-energy protons (e.g., Albert et al., 1998; Selesnick et al., 2016) and the electron-dominated outer belt where electron fluxes vary widely both temporally and spatially (e.g., Reeves et al., 2003; Baker et al., 2014b; Turner et al., 2014). This variability is driven by geomagnetic storm and substorm processes, by changes in the inner magnetospheric

conditions and by wave activity. These processes are initiated by solar wind energy input and disturbances in the solar wind impacting the Earth's magnetosphere (e.g., Daglis et al., 2019). The inner and outer belts are separated by a slot region at about two to three Earth radii characterised by its low flux levels, though it can be flooded by electrons during storms (e.g., Baker et al., 2004).

The Van Allen belts expose spacecraft travelling beyond the low Earth orbit to hazardous radiation (e.g., Feynman and Gabriel, 2000; Horne and Pitchford, 2015; Green et al., 2017; Hands et al., 2018), and the geostationary orbit favored by telecommunication and navigation satellites resides at the outer edge of the outer belt. While inner zone protons impose the most dangerous single effects, prolonged exposure to high energy electrons due to possible sudden enhancements in the highly time-varying outer radiation belt is a significant space weather related threat for satellite operation. The increasingly popular

nanosatellites that often have less shielding available than larger spacecraft are especially vulnerable to the bombardment of energetic particles in the radiation belts.

    The electron fluxes in the outer radiation belt vary on timescales from minutes to days as a result of different acceleration, transport and loss processes. Wave-particle interactions play a key role in the electron dynamics and outer belt response to geomagnetic disturbances (e.g., Thorne, 2010). Forecasting the outer radiation belt dynamics and our understanding of the

competing and coupled belt processes are still incomplete. Detailed studies focusing on how radiation belts respond to different solar wind drivers can shed light on the prompt evolution of the outer belt and improve forecasting models used by the satellite industry.

    Outer belt electrons are usually divided into different populations based on their energy: source (tens of kiloelectron volts), seed (hundreds of kiloelectron volts) and core (megaelectron volts). The highest energy population ($> \sim 3$ MeV) is referred to

as ultrarelativistic electrons. Source and seed electrons can originate from substorm injections, and the source population excite very-low frequency (VLF) chorus waves which, in turn, can progressively accelerate seed electrons to relativistic energies (Jaynes et al., 2015). Chorus waves may also scatter outer belt electrons into the loss-cone (e.g., Bortnik and Thorne, 2007). Another important wave mode changing the outer belt electron fluxes are ultra-low frequency (ULF) waves. ULF Pc5 pulsations (frequency range 2–7 mHz) are generated, e.g., by Kelvin–Helmholtz instability at the magnetopause flanks (Claudepierre

et al., 2008; Wang et al., 2017), by shocks and pressure pulses in the solar wind (Kepko and Spence, 2003; Claudepierre et al., 2010) and by perturbations in the ion foreshock (Hartinger et al., 2013; Wang et al., 2017). ULF Pc5 waves can lead to inward or outward radial diffusion of electrons, resulting in acceleration or losses, respectively (Su et al., 2015; Shprits et al., 2006). On the other hand, electromagnetic ion cyclotron (EMIC) waves, which are generated by temperature anisotropy of ring current protons, contribute to outer belt losses via resonant pitch-angle scattering leading to electron precipitation into the atmosphere

(Usanova et al., 2014; Blum et al., 2019). ULF Pc5 waves can modulate the precipitation, e.g., by lowering the mirror point of electrons (Brito et al., 2012) or by decreasing the minimum energy for resonance with EMIC waves (Zhang et al., 2019). Incoherent plasmaspheric hiss also scatters electrons and is thus important for the formation of the quiet time slot region (e.g., Abel and Thorne, 1998; Jaynes et al., 2014).

    The response of the outer radiation belt to geomagnetic storms has been studied extensively. These studies have considered

the response generally due to storm events (e.g., O'Brien et al., 2001; Reeves et al., 2003; Anderson et al., 2015; Turner et al.,

2015; Moya et al., 2017; Murphy et al., 2018), as well as investigated the significance of the different storm drivers (e.g., Kataoka and Miyoshi, 2006; Hietala et al., 2014; Kilpua et al., 2015, 2019b; Turner et al., 2019). The studies have found that the response depends on both electron energy and radial distance from the Earth (i.e., $L$-shell, see McIlwain, 1961), and that different storm drivers cause distinct responses. The source and seed populations are dominated by enhancement, which tends to occur throughout the outer belt for source electrons and usually at lower $L$-shells for seed electrons, whereas the response of relativistic electrons is more evenly divided between enhancement, depletion and no change events (Turner et al., 2015, 2019). Source and seed populations that have been enhanced due to substorm activity, along with the interplay of chorus waves and electrons at these energies, play a large role in the radiation belt dynamics (Bingham et al., 2018, 2019; Katsavrias et al., 2019a).

One of the most important drivers of geomagnetic activity are interplanetary coronal mass ejections (ICMEs; e.g., Kilpua et al., 2017) that enable effective magnetic reconnection at the magnetopause when their magnetic field has a strong southward component. An ICME that is sufficiently faster than the preceding solar wind will create a shock in front of it, and the turbulent region between the shock front and ICME ejecta is called the sheath region. The shock, sheath and ejecta of an ICME have distinct solar wind properties and magnetospheric impact (see review by Kilpua et al., 2017): Sheaths are turbulent and compressed structures with large-amplitude magnetic field variations and high dynamic pressure, while ejecta are typically characterised by smoothly changing magnetic field direction and low dynamic pressure. The outer belt response to sheaths and ejecta separately and their combination ("full ICME") have been studied, e.g., by Kilpua et al. (2015) and Turner et al. (2019). They found that energetic electrons ($> 1$ MeV) are more likely depleted during geomagnetic storms driven by only sheaths or ejecta, while full ICME events are more likely to result in enhancement at this energy level. Kilpua et al. (2019b) performed a case study of a complex driver consisting of multiple sheaths and ejecta. They found that sheaths were associated with stronger wave activity in the inner magnetosphere than the ejecta.

However, in most previous studies only moderate or stronger geomagnetic storms (*Dst* or *SYM-H* minimum of -50 nT or less) have been considered and the belt response has been computed over long time intervals, usually excluding fluxes in a 24-hour period centred around the *Dst* or *SYM-H* minimum. Our study focused on the immediate outer belt response to sheath regions and considered also weak and non-storm events. Previous studies have shown that large geomagnetic activity is not required for significant changes in the outer radiation belt electron fluxes (Schiller et al., 2014; Anderson et al., 2015; Katsavrias et al., 2015). Furthermore, the radiation belts are an open system that particles enter via injections and exit through losses to the magnetopause and atmosphere. Thus, to account for the total energy budget in the inner magnetosphere, we need to quantify enhancement and losses on timescales shorter than 24 hours. This immediate response is fundamental to distinguish the effects of ICME sheaths and ejecta and critical for enhancing our understanding of the Earth's radiation belt environment.

In this study, we consider the changes in the outer radiation belt electron fluxes by comparing the fluxes from only a few hours before and after the sheath region, as a comparison to the up to a few days intervals used in previous belt response studies. We also comprehensively analyse, for the first time, the geospace response during sheath regions and compare it to the responses during the preceding solar wind and the trailing ejecta. This analysis includes geomagnetic activity indices, subsolar magnetopause and plasmapause locations, energy and $L$-shell dependent outer belt electron fluxes and inner magnetospheric

wave activity (chorus, Pc5, EMIC and hiss). In addition to stronger geomagnetic activity ($SYM\text{-}H_{min} < -50$ nT), our study includes sheaths that caused only a weak geomagnetic storm ($-30$ nT $> SYM\text{-}H_{min} > -50$ nT) or no geomagnetic storm at all ($SYM\text{-}H > -30$ nT).

The paper is organised as follows. Section 2 presents the in situ data sets and the methods employed in our statistical study. We describe an example event and detail our statistical results in Section 3. In Section 4, we conclude our study and discuss future possibilities.

## 2 Data and Methods

### 2.1 Data

We considered 37 CME-driven sheath regions in the Van Allen Probes era since the launch of the spacecraft in August 2012. The events were selected based on the sheath region list compiled by Palmerio et al. (2016) for the period 2012–2015, and for the 2016–2018 interval, we used the sheath list in Kilpua et al. (2019a). We also added three events in 2016 that were identified by visual inspection of solar wind data. The timing of the shocks (i.e., sheath region front boundary) has been taken from the Heliospheric Shock database (http://www.ipshocks.fi) and the ejecta leading edge (i.e., sheath region end boundary) were adjusted to match the boundary between the turbulent and compressed sheath plasma and the ejecta. The typical properties of sheath regions and ejecta, and the challenges in determining the boundary timings, are discussed, e.g., in Richardson and Cane (2010) and Kilpua et al. (2017), and references therein. We included in this study only the cases with well-defined sheath and ejecta boundaries.

The solar wind data were obtained from the Wind spacecraft that monitors solar wind at Lagrangian point L1. We used measurements from the Magnetic Fields Investigation (MFI; Lepping et al., 1995) and Solar Wind Experiment (SWE; Ogilvie et al., 1995) on Wind. The Wind data was time shifted to the bow shock nose. The geomagnetic activity indices were taken from the OMNI database. The Wind and OMNI data were obtained through the NASA Goddard Space Flight Center Coordinated Data Analysis Web (CDAWeb, https://cdaweb.gsfc.nasa.gov/index.html/).

The twin Van Allen Probes (formerly known as the Radiation Belt Storm Probes; denoted RBSP-A and RBSP-B) are on highly elliptical orbits traversing through the inner and outer radiation belts (Mauk et al., 2013). The outer belt electron flux is measured as a function of radial distance and electron energy by the Magnetic Electron Ion Spectrometer (MagEIS; Blake et al., 2013) and Relativistic Electron Proton Telescope (REPT; Baker et al., 2013) in the Energetic Particle, Composition, and Thermal Plasma (ECT; Spence et al., 2013) instrument suite onboard the RBSP spacecraft. MagEIS covers electron energies from 30 keV to 1.5 MeV (source, seed and core populations), while the core and ultrarelativistic electron populations are monitored by REPT, covering energies from 1.8 to 6.3 MeV. In this study, we employed the Level 2 spin-averaged differential electron flux data. For MagEIS electron fluxes, we used only the background corrected data (Claudepierre et al., 2015). The temporal resolution of these data is 11 s. We note that there is considerable variability in the energy scale of the MagEIS energy channels early in the mission up to September 2013 (Boyd et al., 2019). Our study includes 13 events during this period and our results could be slightly affected by these changes. We focused our study to the outer radiation belt between

$L = 2.5$ and $L = 6$. The lower bound was chosen to avoid proton contamination of REPT in the inner belt, and the upper bound was constrained by the Van Allen Probes apogee. The $L$ parameter (McIlwain, 1961), computed using the TS04D magnetic field model (Tsyganenko and Sitnov, 2005), was extracted from the magnetic ephemeris data available on the ECT website (https://rbsp-ect.lanl.gov/).

The very-low frequency (VLF) wave activity in the inner magnetosphere, including chorus waves and plasmaspheric hiss, was obtained from the Electric and Magnetic Field Instrument Suite and Integrated Science (EMFISIS; Kletzing et al., 2013) on the Van Allen Probes. The utilised data product was the Level 2 Waveform Receiver (WFR) diagonal spectral matrix containing the autocorrelations of electric and magnetic field components in 65 frequency bins. The frequency range spans from 2 Hz to 12 kHz, and the spectra are available with a 6 s time cadence. The EMFISIS team also provides electron densities estimated
from the upper hybrid resonance frequency as Level 4 data products (Kurth et al., 2015).

       Additionally, observations of wave activity in the ultra-low frequency (ULF) range were taken from the GOES-15 spacecraft at geostationary orbit ($L \sim 6.6$). The magnetic field data is sampled at 0.512 s by the GOES fluxgate magnetometers (Singer et al., 1996).

## 2.2   Superposed epoch analysis

In superposed epoch analysis, the median of a given parameter is calculated from the data of all events aligned with respect to some reference time (i.e., the zero-epoch time). This technique has been used in various studies to statistically investigate for example solar wind properties, wave activity and electron fluxes (e.g., O'Brien et al., 2001; Kataoka and Miyoshi, 2006; Kilpua et al., 2013, 2015; Hietala et al., 2014; Murphy et al., 2018; Turner et al., 2019). We chose the zero-epoch time to be at the shock and set an additional reference time at the ICME ejecta leading edge (i.e., at the end of the sheath region). We resampled
the data during sheath regions to the same duration. This double epoch analysis allows us to study the general trends in the solar wind parameters and inner magnetospheric activity during driver structures which cover a large range of durations (for similar methods see, e.g., Kilpua et al., 2015; Masías-Meza et al., 2016; Yermolaev et al., 2018). The duration of the studied sheath regions varied widely from 3.0 h to 22.7 h with a standard deviation of 5.3 h. The mean sheath duration was 12.0 h. We resampled the sheath regions to match this mean sheath duration (Kilpua et al., 2013; Hietala et al., 2014). First, the data
during the sheath was rescaled to start at 0 h and end at 12.0 h, and then the data was linearly interpolated to share the same time step in each event. For data that can vary over orders of magnitude (i.e., electron fluxes and wave power), we linearly interpolated the logarithm of these data. We note that some fluctuations can be averaged out when stretching or compressing the sheaths with linear interpolation. However, this should not affect the results significantly as superposed epoch analysis of sheaths with durations close to the mean duration of 12.0 h, ranging from 10 to 14 hours (not shown), presented similar trends
as the full set of events. The superposed epoch analysis was performed for geomagnetic indices, solar wind parameters, inner magnetospheric wave activity, and electron flux in the heart of the outer radiation belt ($L = 3.5$–5).

       We considered wave activity in the very-low and ultra-low frequency ranges in the superposed epoch analysis. Chorus waves appear outside the plasmasphere (where plasma density is approximately $< 50$–$100 \ \mathrm{cm^{-3}}$) in two distinct frequency bands (Burtis and Helliwell, 1969; Koons and Roeder, 1990): the lower band ($0.1$–$0.5 f_{ce}$) and the upper band ($0.5$–$0.8 f_{ce}$), where $f_{ce}$

is the electron cyclotron frequency. Plasmaspheric hiss waves occur inside the plasmasphere in a frequency range from about 100 Hz to $0.1 f_{ce}$. We included a study of hiss waves for completeness, but note that the timescales they operate on outer belt electrons ($> 2$ days; Jaynes et al., 2014) are longer than the sheath durations. We calculated the electron cyclotron frequency $f_{ce}$ based on the TS04D geomagnetic field model (Tsyganenko and Sitnov, 2005). To determine whether the spacecraft was

located inside or outside the plasmasphere at the time of wave measurement, we estimated the plasmapause location with the *AE* index based and magnetic local time dependent model by O'Brien and Moldwin (2003). A plasmapause model was used because the density estimate data is sporadic.

For ultra-low frequency waves, we calculated the wave power spectral density with wavelet analysis from the magnetic field magnitude measured by GOES-15 at geostationary orbit. We calculated the Pc5 wave power in the range from 2.5 to 10 min

(2–7 mHz) and the EMIC wave power in the range from 0.2 to 10 s (0.1–5 Hz) which corresponds to the range of Pc1 and Pc2 pulsations as given by Jacobs et al. (1964). The power spectral densities were averaged in the given frequency ranges of the wave modes to obtain the wave power data for the superposed epoch analysis, and resampling was performed after this averaging. We note that GOES measurements taken at geostationary orbit might not always reflect the ULF Pc5 and EMIC wave activity at the position of the Van Allen Probes (Engebretson et al., 2018; Georgiou et al., 2018).

For electron flux in the superposed epoch analysis, we considered the median flux in the heart of the outer belt at $L = 3.5$–5. The MagEIS and REPT electron flux measurements were binned in time ($\Delta t = 4$ h) and $L$-shell ($\Delta L = 0.1$) to combine the data from the two spacecraft. The 4-hour cadence was chosen to reduce the effect of the Van Allen Probes orbits in order to minimise the data gaps in the binned flux data during all events. We note that a 4-hour cadence leaves us with only four data points during the sheath, but we are here mostly interested in the overall trend during the events, which is similar at higher

time resolutions. We selected four energy channels to represent the source (54 keV), seed (346 keV), core (1,064 keV) and ultrarelativistic (4.2 MeV) populations. We also calculated the mean electron flux at $L = 3.5$–5 with the same time and L-shell bins (not shown), and note that the trends are very similar to the median values.

In addition to investigating the median sheath properties of all 37 events, we divided the events based on the level of associated geomagnetic activity inferred from the *SYM-H* index. The *SYM-H* index (Iyemori, 1990; Iyemori and Rao, 1996)

is derived from perturbations in the horizontal component ($H$) of the geomagnetic field that is affected by changes in the ring current. The *SYM-H* index is essentially the same as the hourly *Dst* index but with a higher time resolution of one minute, and it is also more sensitive to substorm activity. The strength of a geomagnetic storm is usually characterised with the minimum *Dst* value, where the thresholds for small, moderate and intense storms are $-30$ nT, $-50$ nT and $-100$ nT, respectively (Gonzalez et al., 1994). In previous studies, events have typically been divided with the threshold of $-50$ nT, or only these moderate or

larger storms are considered (e.g., O'Brien et al., 2001; Reeves et al., 2003; Kilpua et al., 2015; Lugaz et al., 2016; Turner et al., 2015, 2019). However, due to the relatively low number of well-defined sheath events during Van Allen Probes measurements, only nine events out of our total 37 events have a *SYM-H* minimum below $-50$ nT during the sheath region or two hours after. Therefore, we set the threshold to $-30$ nT to obtain a statistically adequate subset of 17 geoeffective events. The interval where we took the minimum was extended two hours after the sheath to accommodate for lag in the ring current response. Note that

the geomagnetic disturbance of the ICME ejecta was not considered.

## 2.3 Electron flux response

We binned the MagEIS and REPT spin-averaged electron flux data from both spacecraft in 0.1 $L$-shell bins and 1-hour time bins, differing from the superposed epoch analysis in order to have a higher time resolution. Based on the methodology of Reeves et al. (2003) and Turner et al. (2015, 2019), we define the outer belt electron response ($R$) as the ratio of post-event flux to pre-event flux. The pre-event flux was obtained by averaging the electron flux over a 6-hour interval before the sheath region, and post-event flux by averaging over six hours after the sheath region. The response parameter $R$ was computed for each considered electron energy and $L$-shell bin. The response was categorised as *enhancement* when the post-event flux had increased by over a factor of 2 as compared to the pre-event flux ($R > 2$), *depletion* when it had decreased by over a factor of 2 ($R < 0.5$) and *no change* when the flux level had not changed significantly ($0.5 \leq R \leq 2$).

In previous studies (Reeves et al., 2003; Turner et al., 2015, 2019), the pre- and post-event fluxes were defined as the maximum flux from $> 12$ h up to a few days before and after the event, since the outer radiation belt response to entire geomagnetic storms was examined. These studies also excluded the 24-hour period during the storm. In the current study, we use the mean flux values close to the sheath region as we focus on the outer belt response to the sheath region only, and all sheaths did not generate geomagnetic storms (in 20 out of 37 events the *SYM-H* index does not drop below $-30$ nT). A post-sheath maximum flux value is not meaningful as it would be embedded in the ICME ejecta and subject to possible fluctuations driven by the ejecta. The 6-hour averaging period aims to capture the changes generated by the sheath while excluding the main response to the ejecta, which is expected to occur later (mean duration of the ejecta was 28.4 h with a standard deviation of 11.1 h).

## 3 Results

### 3.1 Example event on 7 February 2014

Figure 1 shows the solar wind conditions and geomagnetic indices during 7–9 February 2014, when an interplanetary coronal mass ejection (ICME) driving a sheath region impacted the Earth. The shock (first red vertical line in Figure 1) was identified as an abrupt and simultaneous increase in the magnetic field and solar wind speed, as well as a small increase in dynamic pressure. Both the sheath and ICME were relatively slow ($\approx 400$–$450$ km s$^{-1}$). The shock was also quite weak, as the speed jump across the shock was about $100$ km s$^{-1}$. The sheath was characterised by fluctuating magnetic field and variable dynamic pressure, which had high values ($\approx 20$ nPa) in the trailing half of the sheath. The ejecta had smoother field and clearly lower dynamic pressure. This ICME is included in the Richardson and Cane ICME list (http://www.srl.caltech.edu/ACE/ASC/DATA/level3/icmetable2.htm; Richardson and Cane, 2010) and is reported there as a "magnetic cloud", i.e., the event shows signatures of a magnetic flux rope. This is because the magnetic field components (Figure 1b) show some organised rotation during the ICME and the north-south magnetic field component ($B_Z$) rotates from north to south. In the sheath, the field was predominantly northward.

The event was only mildly geoeffective (Figure 1f,g). The *SYM-H* index dropped to $-29$ nT in the middle of the sheath (and briefly below $-30$ nT an hour after the sheath ended) and the ICME caused only a weak storm. The substorm activity was also weak (but quite continuous) as shown by the *AL* index.

Despite the low geoeffectiveness of both the sheath and ejecta, there were clear changes in the outer radiation belt electron fluxes at source, seed, core and ultrarelativistic energies as shown in Figure 2. Note that for this particular event, background corrected fluxes are not available for $L > 3$ at source energies. Before the shock arrived, the outer belt showed typical undisturbed conditions (e.g., Reeves et al., 2016) with the seed and core electron fluxes being higher at the highest $L$-shells. The ultrarelativistic electrons in turn peaked at $L \sim 4$. After the shock arrival, the fluxes increased at seed and higher energies. The most distinct increase was detected at ultrarelativistic (4.2 MeV) energies. The fluxes at seed, core and ultrarelativistic energies also widened towards lower $L$-shells during the sheath. The flux of the source population (54 keV) increased significantly at the end of the sheath. At higher energies, on the other hand, the flux was depleted near the sheath–ejecta boundary. Interestingly, after the sheath ultrarelativistic electron fluxes were enhanced already in the front part of the ICME ejecta, while seed and core electron fluxes increased clearly only near the middle of the ejecta.

The wave activity in the inner magnetosphere during the event is illustrated by Figure 3, which shows the wave power spectral density of both very-low and ultra-low frequency waves as measured by RBSP-B and GOES-15, respectively. Some chorus activity (Figure 3a) appeared immediately after the shock, and it was enhanced in the latter half of the sheath region. Chorus activity persisted during the ejecta. The chorus waves might have caused some acceleration, e.g., the enhancement of 1 MeV electrons during the ejecta, but the waves would not have yet acted long enough to cause the enhancement of ultrarelativistic electrons during the sheath (e.g., Jaynes et al., 2015).

The ULF wave power in the Pc5 and EMIC ranges was elevated during the sheath (Figure 3f). The widening of electron fluxes towards lower $L$-shells could thus be a result of inward radial transport by ULF waves (e.g., Turner et al., 2013; Jaynes et al., 2018). EMIC waves can also be responsible for the loss of relativistic electrons (Usanova et al., 2014). The subsolar magnetopause was located at about 12.7 $R_E$ before the shock arrival, according to the Shue et al. (1998) model (Figure 1e). The shock pushed the magnetopause nose inward and it was located closest to Earth ($\approx 7.5\ R_E$) at the end of the sheath when depletion occurred. At the ejecta leading edge, the magnetosphere started to recover and the subsolar magnetopause stayed at about 10 $R_E$ during the ejecta. Therefore, losses at the magnetopause (i.e., magnetopause shadowing) could be the main driver of depletion, possibly coupled with outward transport by ULF waves (Turner et al., 2012). During the ejecta, chorus waves continued, but ULF Pc5 and in particular EMIC wave activity weakened. The enhancement of seed and core fluxes in the ejecta was thus likely associated with continued chorus activity and possibly also with inward transport by ULF Pc5 waves.

## 3.2 Statistics of 37 sheath events

Statistics from the superposed epoch analysis of 37 events with sheath regions are presented in Figure 4. Results are also shown for 10 hours of solar wind before the shock and for one day of the ICME ejecta after the sheath (note that unlike the sheath regions, the ejecta were not resampled). The results in panels (a)–(d) show the general characteristics of sheath regions (e.g., Kilpua et al., 2017, 2019a): lower magnetic field magnitude than in the ejecta (but about twice as strong as in the quiet solar

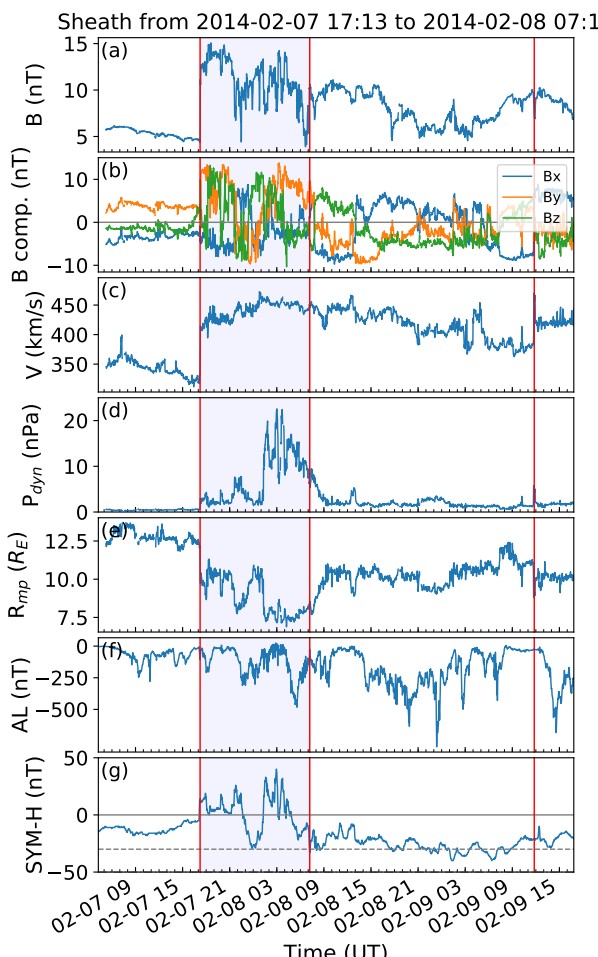

**Figure 1.** (a) Magnetic field magnitude, (b) magnetic field components in the geocentric solar magnetospheric coordinate system, (c) solar wind speed, (d) solar wind dynamic pressure, (e) subsolar magnetopause location from the Shue et al. (1998) model, (f) *AL* index and (g) *SYM-H* index. The red vertical lines indicate the shock, ICME ejecta leading edge and ejecta trailing edge in UT (universal time). The shaded area marks the sheath interval. The dashed line in the last panel indicates $SYM\text{-}H = -30\,\mathrm{nT}$.

wind), elevated dynamic pressure (as well as solar wind density) as compared to quiet solar wind conditions and the ejecta, and contracted magnetopause nose due to the high-dynamic pressure sheath. The *SYM-H* index usually has a positive peak at the shock (corresponding to the storm sudden commencement and initial phase/sudden impulse), and then it gradually decreases during the sheath (Figure 4e). However, on average, the main geomagnetic storm impact is imposed by the ejecta. In 17 events, the *SYM-H* index dropped below $-30\,\mathrm{nT}$ (weak storm) during the sheath or two hours after, and it dropped below $-50\,\mathrm{nT}$ (moderate storm) only in nine events. On average, only weak substorm activity is evidenced by the *AL* index during the sheath region and ejecta (Figure 4f).

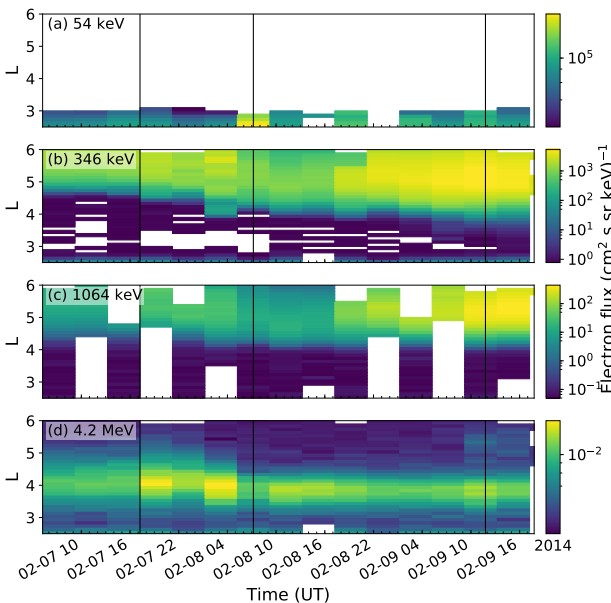

**Figure 2.** The spin-averaged electron fluxes measured by MagEIS at (a) 54 keV (source), (b) 346 keV (seed) and (c) 1,064 keV (core) and by REPT at (d) 4.2 MeV (ultrarelativistic). The data are combined from both Van Allen Probes and are binned by 4 hours in time and 0.1 in $L$-shell. The vertical lines mark the sheath region and ICME ejecta intervals.

Panels (g)–(l) of Figure 4 show the statistics of different wave modes in the inner magnetosphere during the selected events. ULF Pc5 wave power peaks in the sheath, showing a growing trend from the shock towards the end of the sheath region (Figure 4g). The mean of median Pc5 wave power in the sheath is about $10^2$ nT$^2$Hz$^{-1}$ as measured by GOES-15 at $L \sim 6.6$, which is one order of magnitude larger than during quiet solar wind. The wave power of Pc5 waves declines during the ejecta.
The EMIC wave power is also larger in the sheath than during the ejecta and quiet solar wind, with a median wave power of about $10^{-2.5}$ nT$^2$Hz$^{-1}$ (Figure 4h). The median EMIC wave power quickly decreases to the pre-event level of about $10^{-3}$ nT$^2$Hz$^{-1}$ in the ejecta.

The main power of chorus waves is in the lower band, where the order of magnitude during the sheath is $10^{-9}$ nT$^2$Hz$^{-1}$ (Figure 4j). The median wave power of upper band chorus is an order of magnitude lower, but in a quarter of the cases the
10 power can reach values comparable with lower band waves as shown by the upper quartile curve (Figure 4k). The chorus wave power is very similar in the sheath and the ejecta, and it is also on average only about four times higher during the sheath than during the pre-event conditions. The chorus wave power increases gradually for a few hours before the shock arrival. This could be driven by the very weak substorm activity in front of the ICME event. Plasmaspheric hiss is not affected by the sheath or ejecta, and its median wave power remains at about $10^{-8}$ nT$^2$Hz$^{-1}$ throughout the event (Figure 4l).
The behaviour of the median electron fluxes in the heart of the outer belt ($L = 3.5$–5) is shown in panels (m)–(p) of Figure 4. The flux of the source population increases during the sheath region and it stays around a constant level during the ejecta.

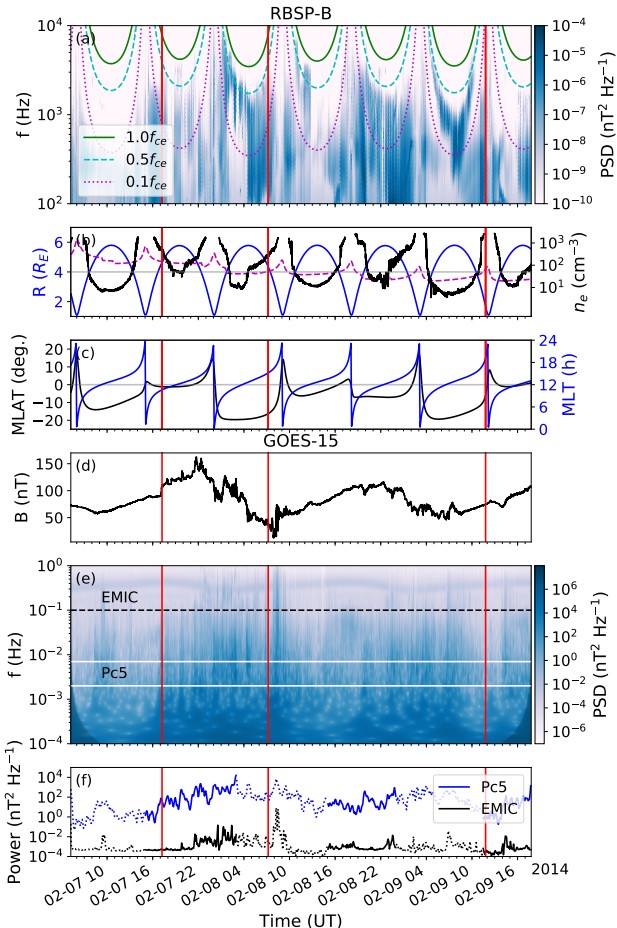

**Figure 3.** Very-low and ultra-low frequency (VLF and ULF) wave activity. (a) Power spectral density of VLF waves from RBSP-B/EMFISIS. The curves indicate different values of the equatorial gyrofrequency $f_{ce}$ calculated from the TS04D geomagnetic field model. Chorus waves have frequencies $> 0.1 f_{ce}$ outside the plasmasphere, and plasmaspheric hiss is present at lower frequencies. (b) TS04D model spacecraft radial location in blue with the model plasmapause location (O'Brien and Moldwin, 2003) shown as a dashed magenta line (left axis), and estimated electron density (right axis). The horizontal line at $50 \text{ cm}^{-3}$ illustrates another estimate of the plasmapause location. (c) TS04D model magnetic latitude (left axis) and magnetic local time (right axis). (d) Magnitude of the magnetic field as measured by GOES-15. (e) Power spectral density of ULF waves from wavelet analysis of the GOES-15 magnetometer measurements. The shaded areas mark the cone of influence. The white horizontal lines indicate the range of Pc5 pulsations, and the dashed horizontal line indicates the lower boundary of the EMIC range. (f) Wave power of ULF Pc5 and EMIC waves. Solid and dotted lines indicated when GOES-15 was on the dayside and nightside, respectively. The red vertical lines indicate the sheath and ICME ejecta intervals.

Comparison of the pre-sheath to post-sheath fluxes shows that the median response of 54 keV electrons is an enhancement by a factor of 5. For the seed population, the flux is slightly enhanced at the shock but, on average, the flux remains unaffected in the sheath. However, the flux suddenly increases after the sheath ends and continues to be enhanced in the ejecta. The 346 keV

electron median flux increases by a factor of about 25. At $\mathrm{MeV}$ energies, the flux in the heart of the outer belt remains mostly unchanged when considering the median response. The flux in the ejecta is less than a factor of 2 larger than before the sheath based on the medians for both $1{,}064\ \mathrm{keV}$ and $4.2\ \mathrm{MeV}$. However, the upper quartile of the core population at $1{,}064\ \mathrm{keV}$ shows a slight increase and the lower quartile a slight decrease, evidencing that these opposite responses are averaged out in

the median. Both the upper and lower quartiles for $4.2\ \mathrm{MeV}$ electrons indicate a slight decrease in the flux.

Superposed epoch analysis of the plasmapause location from an *AE* index based model (O'Brien and Moldwin, 2003) is shown in Figure 5 both as independent on magnetic local time (MLT) throughout the event and as MLT-dependent for the pre-event time ($-6$ hours from shock), sheath region ($+6$ hours) and ejecta ($+18$ hours). One event in 2018 was excluded in the analysis due to the *AE* index data not being available. The MLT-dependence of the model shows that the plasmapause is

closer to the Earth on the dayside and further away in the nightside during both quiet and disturbed times. In the preceding solar wind, the plasmapause is located at about 5 $R_E$. During the sheath, the plasmasphere moves earthward, and it moves even further earthward during the ejecta. The variation is consistent with the general *AL* levels in the preceding solar wind, sheath and ejecta (the *AE* index should roughly follow *AL* behaviour). At noon MLT, the median plasmapause location moves from about 4.4 $R_E$ during the quiet solar wind conditions to 3.6 $R_E$ in the middle of the sheath, and 6 hours after the sheath region

(18 hours after the shock) the median distance has decreased to 3.3 $R_E$.

The electron flux response of the whole outer radiation belt for a wider selection of energies than in the superposed epoch analysis is shown in Figure 6, where the response is divided to the three categories of enhancement, depletion and no change. The source population at $L > 3.5$ is enhanced in 80% of the cases, and practically never depleted. Closer to the inner boundary of the outer belt, no change events are very common at all energies. Electrons at seed energies are enhanced in about half of the

cases throughout the belt, with a higher possibility for enhancement in the heart of the outer belt. In a small subset ($< 15\%$) of the seed electrons, depletion occurs near $L \sim 3$. Depletion is most common in the $\sim 1$–$3\ \mathrm{MeV}$ core population at high $L$-shells ($L > 4.5$). At lower $L$-shells, core electron flux is enhanced at most in 10% of the cases, and in a major fraction of the events ($> 60\%$) the core electron fluxes do not change significantly below $L \sim 4.5$.

Interestingly, a feature in the outer belt response is that the depletion progresses to lower energies when $L$ increases. At

$L \sim 4.5$ depletion dominates only at $> 2\ \mathrm{MeV}$ energies, while at $L \sim 6$ it has reached down to seed energies at around $500\ \mathrm{keV}$.

### 3.3   Impact of geoeffectiveness

Dividing the studied 37 sheath events based on the geomagnetic disturbance they cause, inferred from the *SYM-H* index ($\leq -30\ \mathrm{nT}$ for geoeffective events) during the sheath region and two hours after it, we found a different response in the outer radiation belt. The superposed epoch analysis results presented in Figures 7 and 8 show that geoeffective events are associated

with larger dynamic pressure and magnetic field magnitude in the sheath and tend to have higher speeds. Geoeffective sheaths are also accompanied with strongly geoeffective ejecta more often than sheaths where the *SYM-H* index remains close to $0\ \mathrm{nT}$. Geoeffective sheaths also have larger positive *SYM-H* peaks at the shock, likely due to their tendency for high dynamic pressure, and as expected, substorm activity is greater during geoeffective events as evidenced by the *AL* index.

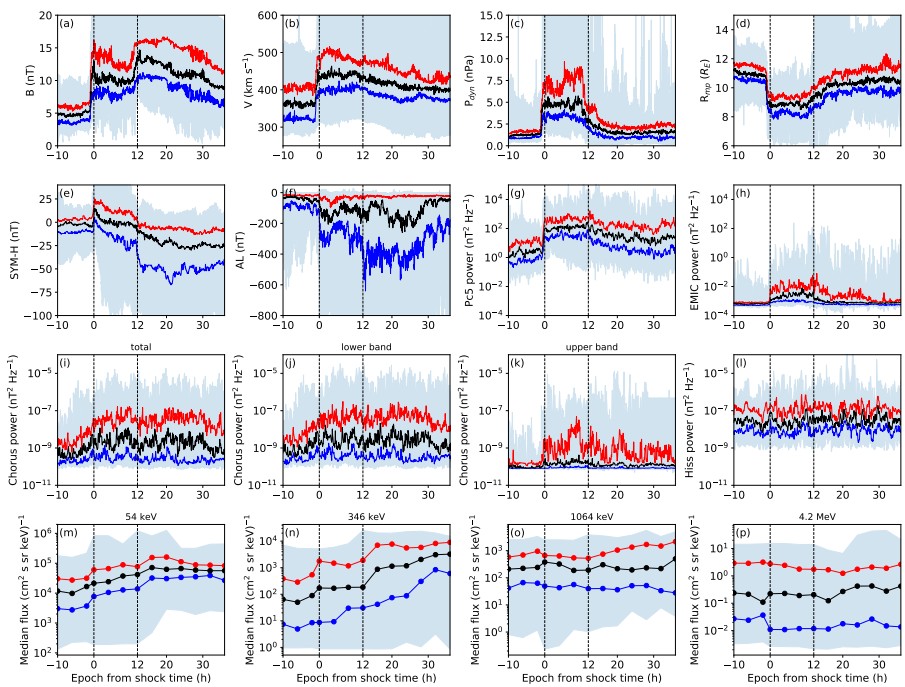

**Figure 4.** Results of superposed epoch analysis for solar wind data, geomagnetic indices, VLF and ULF wave powers and median electron fluxes in the heart of the outer radiation belt ($L = 3.5$–$5$). The black curves show the medians, and the red and blue curves show the upper and lower quartiles, respectively. The shaded area indicates the full range of data from all events. Panels show (a) interplanetary magnetic field magnitude, (b) solar wind speed, (c) solar wind dynamic pressure, (d) subsolar magnetopause location from the Shue et al. (1998) model, (e) *SYM-H* index and (f) *AL* index. Panels show wave activity as (g–h) ULF Pc5 and EMIC wave power, (i–k) total, lower band and upper band chorus wave power, and (l) plasmaspheric hiss wave power. The median electron fluxes are shown in panels (m–p) at source (54 keV), seed (346 keV), core (1,064 keV) and ultrarelativistic (4.2 MeV) energies, respectively. The wave power of ULF waves (Pc5 and EMIC) was computed from GOES-15 measurements, whereas VLF wave power (chorus and hiss) was obtained from RBSP-A and RBSP-B with the plasmapause location taken into account (using the model by O'Brien and Moldwin, 2003).

Wave power of Pc5, EMIC and chorus waves is higher by a factor of about 6 in geoeffective sheaths as compared to non-geoeffective ones. In geoeffective events, the jump in wave power at the shock is larger in all considered wave modes. For example, the median Pc5 wave power is about 50 times higher during the sheath than before the shock arrival in geoeffective events, whereas in non-geoffective cases it is only about 20 times higher. Pc5 wave power also gradually decreases in geoef-

5  fective ejecta, but during less effective ejecta it remains at an approximately constant level that is lower than the median power in the sheath, though the wave power has a slightly increasing trend near the end of the considered period. The median EMIC wave power behaves similarly between the two groups of events.

While the median chorus wave power in geoeffective events increases, on average, by an order of magnitude from pre-event conditions to the sheath region, the chorus activity does not significantly differ between the sheath and ejecta, where it is about

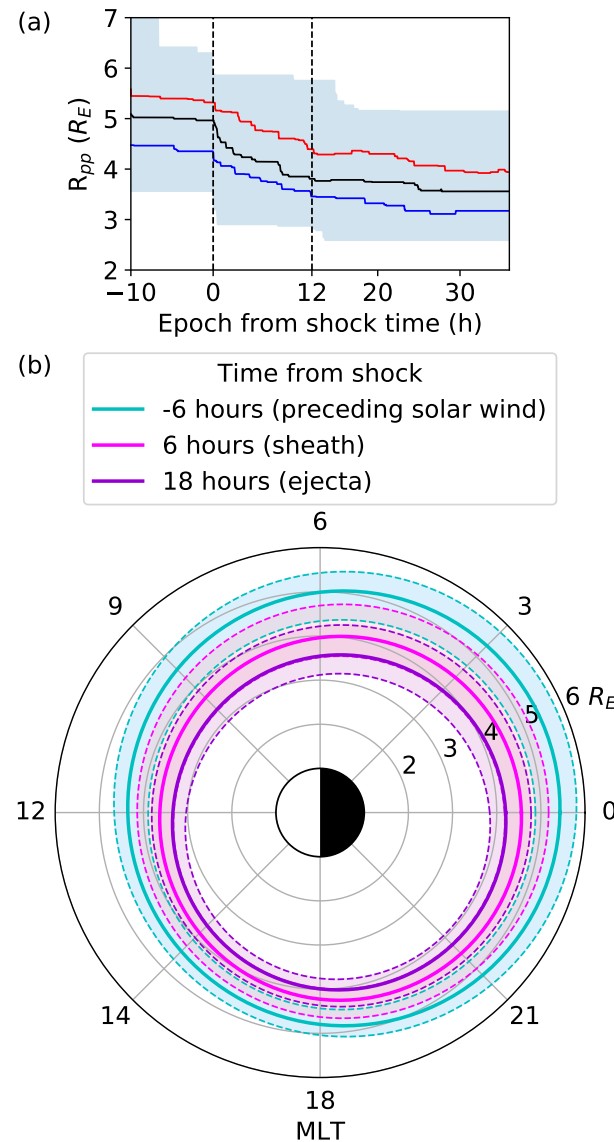

**Figure 5.** Results of superposed epoch analysis for the model plasmapause location (O'Brien and Moldwin, 2003) for the non-MLT dependent case and as a function of MLT. (a) The non-MLT dependent results are presented in the same format as results in Figure 4. (b) The MLT-dependent plasmapause is shown at six hours before (cyan), six hours after (magenta) and 18 hours after (violet) the epoch time at the shock, sampling the pre-event, sheath and ejecta regions, respectively. The medians are shown in solid lines, while the upper and lower quartiles are indicated with the dashed lines and shaded area.

$10^{-8}$ nT$^2$Hz$^{-1}$ for lower band and $10^{-9}$ nT$^2$Hz$^{-1}$ for upper band chorus. We also note that in a quarter of the geoeffective cases the upper band chorus wave power is significantly enhanced at the shock (Figure 7k). Non-geoeffective sheaths that are

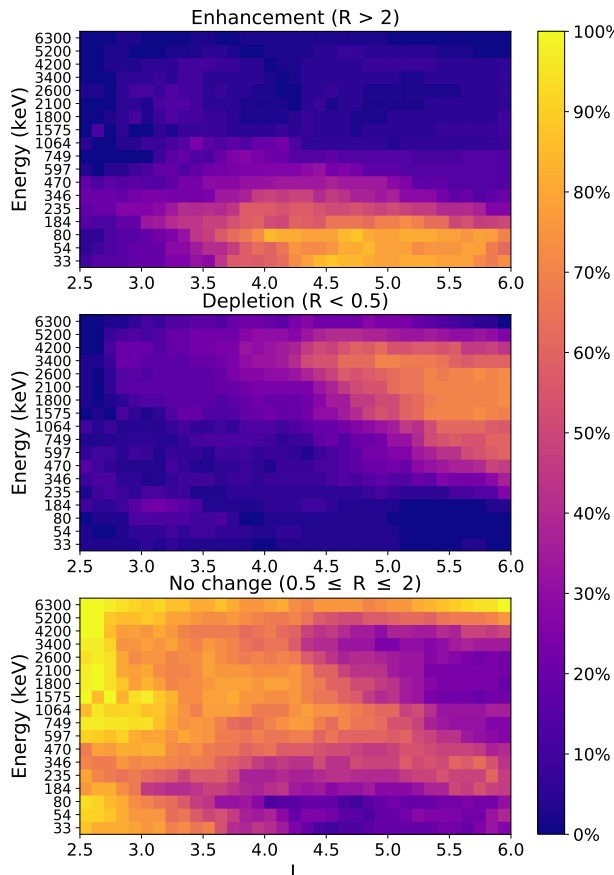

**Figure 6.** Percentage of the sheath events causing enhancement (top), depletion (middle) or no change (bottom) in the outer radiation belt electron fluxes as a function of electron energy and $L$-shell (0.1 bins). The sum of percentages from all three panels for a given energy and $L$-shell bin is 100%.

associated with modest substorm activity drive chorus waves only in about a quarter of the events, and the median chorus wave power remains roughly at the pre-event level throughout the ICME ($10^{-9}$ nT$^2$Hz$^{-1}$ for lower band and $10^{-10}$ nT$^2$Hz$^{-1}$ for upper band chorus), as opposed to geoeffective events where substorm injections during the ICME excite stronger chorus activity. The median wave power of plasmaspheric hiss is on average twice as high during geoeffective events than during non-geoeffective events.

The median fluxes in the heart of the outer belt experience enhancement at all considered energies in geoeffective events. The strongest increase occurs in the seed population, whose median flux increases by almost two orders of magnitude. During the sheath, the flux gradually increases at source and seed energies, while it decreases and is the lowest during the sheath at MeV energies.

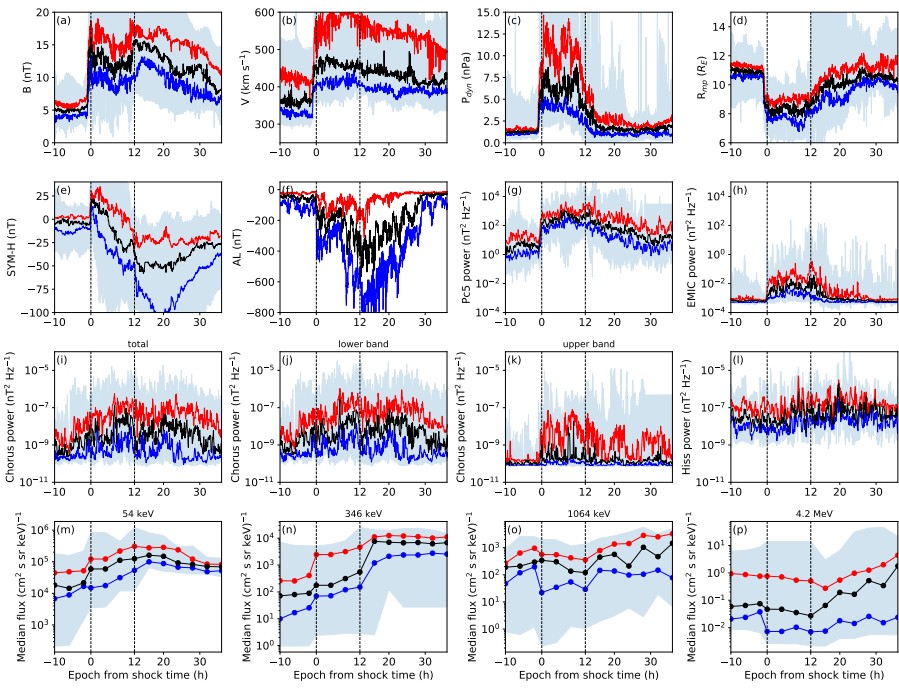

**Figure 7.** Same as Figure 4 but for geoeffective sheath events where the *SYM-H* index has a minimum of $-30$ nT or smaller during the sheath or two hours after it. The total number of geoeffective events is 17.

For non-geoeffective events, the source and seed populations are enhanced, but flux at core energies does not significantly change and the ultrarelativistic population is depleted. This differs from the geoeffective case where enhancement occurred at all four energies. The median electron flux at 54 keV increases throughout the event, but the seed population at 346 keV remains on the same level during the sheath after an initial large increase at the shock and the flux begins to increase again only after a few hours in the ejecta. The 1,064 keV electron flux is slightly enhanced during the sheath before the depletion, while electron losses at 4.2 MeV energies take place already at the shock. The change in median fluxes is also lower than in geoeffective events, with the largest change being an increase by a factor of 20 at seed energies in non-geoeffective events.

Again, the outer belt response as a function of $L$-shell and a wider range of electron energies is considered and the results are shown separately for geoeffective and non-geoeffective events in Figure 9. It is immediately evident that for geoeffective sheaths, enhancement events are more common at all energies and $L$-shells, and the source and seed populations are practically always enhanced in the heart of the outer belt ($L = 3.5$–$5$). However, deviating from the superposed epoch analysis results, $> \mathrm{MeV}$ electrons experience depletion more frequently in geoeffective events throughout the outer belt. In non-geoeffective events depletion begins to dominate the core population response only at around $L > 5$. Virtually all non-geoeffective events result in no significant change at low $L$-shells ($L < 4.5$) at almost all energies, while flux enhancements take place practically only at source energies and are limited to $L > 4$.

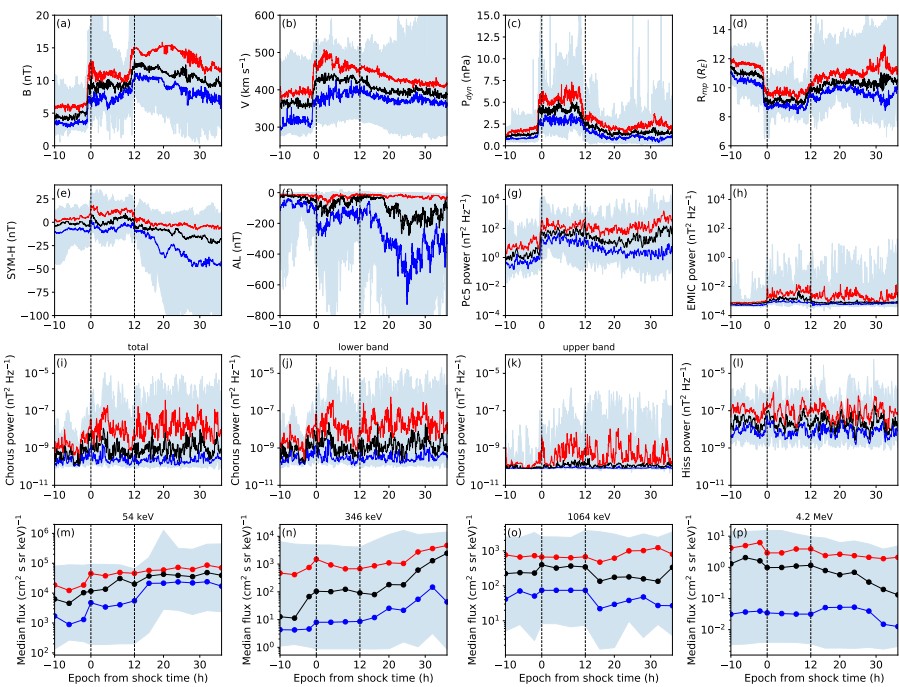

**Figure 8.** Same as Figure 4 but for non-geoeffective sheath events where the *SYM-H* index remains above −30 nT during the sheath and two hours after it. The total number of non-geoeffective events is 20.

## 4 Discussion and Conclusions

In this paper, we studied statistically the inner magnetospheric wave activity as well as the energy and *L*-dependent outer radiation belt electron flux response during ICME-driven sheath regions. Our study included 37 sheaths during the Van Allen Probes era (2012–2018).

We found that turbulent sheath regions preceding ICMEs caused significant changes in the outer radiation belt electron fluxes. While the response was the most dramatic for geoeffective sheaths, we emphasise that these changes also occurred during the sheaths that caused only a weak geomagnetic storm or that were not geoeffective at all in terms of their *SYM-H* response. These results are consistent with previous findings that have reported clear responses during small geomagnetic storms (Anderson et al., 2015) and also during non-geoeffective sheaths in case studies (e.g., Alves et al., 2016; Kilpua et al.,
2019b). The ejecta in our data set had larger *SYM-H* response than the sheath regions.

    Our analysis showed that the inner magnetospheric wave activity was clearly enhanced in the sheath when compared to the preceding solar wind; Pc5 wave power was enhanced by one order of magnitude and EMIC and chorus wave power was four times higher than in the preceding solar wind. We also found that ULF Pc5 and EMIC wave power were larger in the sheath than in the following ejecta. This is in agreement with a previous case study by Kilpua et al. (2019b). As discussed in Kilpua
et al. (2019b), the ULF enhancement is likely due to higher and variable dynamic pressure and more turbulent variations of

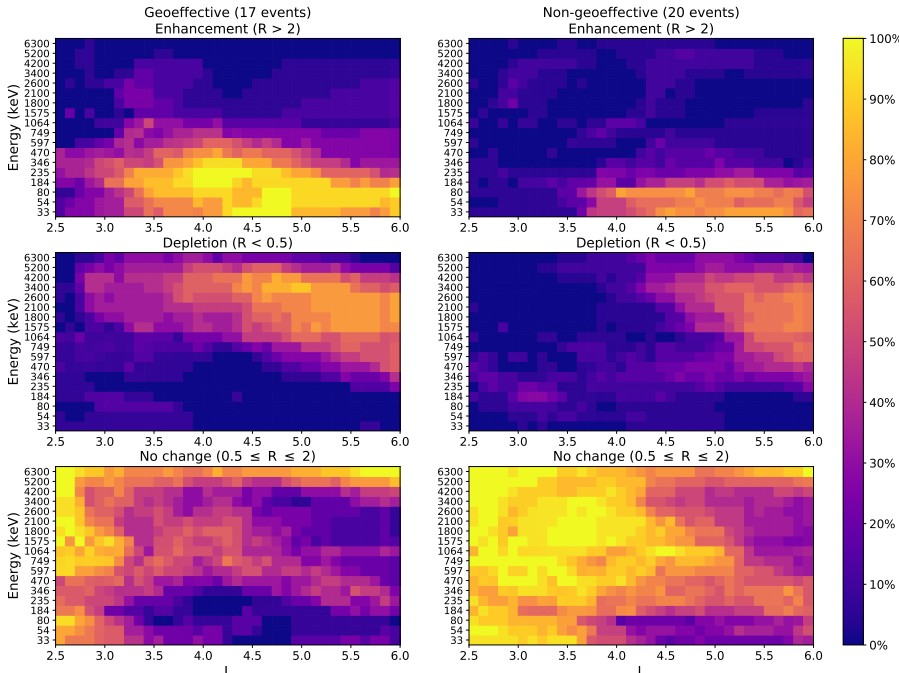

**Figure 9.** Percentages of sheath events causing enhancement, depletion or no change similar to Figure 6 with events divided based on their geoeffectiveness. Left column: Geoeffective events with a *SYM-H* minimum of $-30$ nT or below. Right column: Non-geoeffective events where $SYM\text{-}H > -30$ nT.

the magnetic field in the sheaths than in the ejecta. We also note that in the solar wind, as opposed to the magnetosphere as investigated here, sheaths have in general a clearly higher level of ULF Pc5 wave power than the ejecta and the preceding solar wind (Kilpua et al., 2013; Hietala et al., 2014). High ULF Pc5 wave power in sheaths can enhance the growth rate of chorus waves (e.g., Coroniti and Kennel, 1970). However, chorus and plasmaspheric hiss wave power had in turn more similar
levels in the sheath and ejecta. Chorus waves are excited by substorm injected electrons. Despite the clearly stronger *SYM-H* response during the ICME ejecta, substorm activity evidenced by the *AL* index was comparable during the sheath and ejecta, except during about a quarter of the cases. Consequently, the chorus activity did not significantly change during the events.

     In previous studies, sheath response has been investigated statistically at geostationary orbit (e.g., Hietala et al., 2014; Kilpua et al., 2015) and with radially resolved Van Allen Probes data over several days time periods (e.g., Turner et al., 2015,
2019). In this work, we detailed more precisely the more immediate sheath response over wide $L$-shell and energy ranges. We found that sheaths deplete relativistic MeV electrons at higher $L$-shells (down to about $L \sim 4.5$). The results showed that enhancements at ultrarelativistic energies are rare, which is in agreement with a previous study by Zhao et al. (2019), who found few enhancement events of ultrarelativistic electrons during weak geomagnetic activity ($Dst_{min} > -50$ nT) during the Van Allen Probes era. We further showed that the highest energy electrons ($>\sim 4$ MeV) throughout the outer belt and 1–
4 MeV electrons in the inner part of the outer belt are mostly unchanged during the sheath passage. The source electrons (tens

of keV) were in turn enhanced throughout most of the outer belt during the sheaths despite their quite mild geoeffectiveness. In about half of the cases, seed electrons (hundreds of keV) were enhanced in the heart of the outer belt, while more energetic seed electrons ($> 500$ keV) depleted in about half of the cases at high $L$-shells. Additionally, our example event showed that even weakly geoeffective sheaths can in some cases result in clear outer radiation belt response up to ultrarelativistic energies.

Since the sheaths cause enhancements of source and seed electrons but mostly depletion of the most energetic seed electrons ($> 500$ keV) and the core population, they cannot, statistically, produce the so-called killer electrons ($> 1$–2 MeV) under the studied timescales.

The results described above agree on a general level with the results of ICME sheath impacts presented by Turner et al. (2019), who only considered events that caused a geomagnetic storm with a *SYM-H* minimum below $-50$ nT. Therefore, we

compare their results to our results for geoeffective events only (*SYM-H* minimum $\leq -30$ nT). Turner et al. (2019) found that seed electrons are enhanced more often than the source population and most enhancements occur at $L < 4$, while our study revealed somewhat opposite results. The source and seed populations in our case are equally likely enhanced and most enhancement take place at $L > 3.5$. On the other hand, Turner et al. (2019) found that depletion of MeV electrons was as likely throughout the outer belt, whereas we show that immediate depletion is more restricted to higher $L$-shells. The different results

between our study and the one by Turner et al. (2019) are most likely attributed to the difference in the time intervals considered in these studies. We investigated the immediate sheath response in six hours before and after the sheath, while Turner et al. (2019) considered 72-hour periods 12 hours before and after the *SYM-H* minimum. Additionally, Turner et al. (2019) included only moderate or stronger storms and used maximum flux values to calculate the response, whereas we used the median fluxes.

The immediate response to the sheath has a clear energy and $L$-shell dependence. High-energy electrons typically cannot

access low $L$-shells $<\sim 4$ except during strong magnetic storms and very strong solar wind drivers (e.g., Baker et al., 2014a; Reeves et al., 2016), however, ultrarelativistic electrons can reach lower $L$-shells also during weak storms via inward radial diffusion (e.g., Zhao et al., 2018; Katsavrias et al., 2019b). At low $L$-shells ($L < 3.5$), the high percentage of no change events at lower energies ($< 300$ keV) is a result of the unaffected inner radiation belt population. At larger energies, no change events at $L < 3.5$ are likely due to the typically weakly populated slot region. At high $L$-shells ($L > 5$), the electron fluxes at seed

energies do not change much as substorm injections effectively replenish the population (e.g., Turner et al., 2019). One distinct feature we found was the clear energy and $L$-shells dependence in the losses (Figure 6). Depletion becomes more likely when energy and $L$-shell increase, but also extends to lower energies with increasing radial distance. Such dependence was not found in previous sheath response studies (e.g., Turner et al., 2019). We suggest that this energy and $L$-shell dependent depletion can be explained by energy-dependent wave-particle interactions contributing significantly to electron losses in the heart of the

outer belt, while at larger radial distances all energies are depleted equally via magnetopause shadowing, possibly enhanced by outward radial diffusion by ULF Pc5 waves.

We also found clear differences in the wave activity and energetic electron response between geoeffective (*SYM-H* minimum $\leq -30$ nT) and non-geoeffective sheaths: wave activity is higher during geoeffective events, and enhancement of the source and seed populations and depletion of the core population are more common. In addition, significant response takes place also

at lower $L$-shells for all energies during geoeffective events (similiar to the results presented in Turner et al., 2019), while

non-geoeffective events usually cause significant changes only at $L > 4$. This can be attributed to geoeffective sheaths having tendency for larger dynamic pressure, stronger ring current (*SYM-H*) and substorm activity (*AL*). Consequently, they show strong seed energy enhancement due to substorms, while $\mathrm{MeV}$ fluxes are depleted more often due to stronger magnetopause shadowing and possible EMIC wave scattering.

The results in this paper roughly agree with the general conclusions of phase space density studies. Using phase space density analysis of the electron fluxes, Reeves et al. (2013) showed that local acceleration, i.e., energisation via wave-particle interactions, dominated in the heart of the outer belt during an intense geomagnetic storm. Turner et al. (2013, 2014) showed in statistical and case studies that outer belt enhancements during geomagnetic storms are associated with local acceleration via chorus waves. Prompt depletion is consistent with magnetopause shadowing and enhanced outward radial transport, and pitch-angle scattering by EMIC waves leads to precipitation loss (e.g., Turner et al., 2013, 2014).

In this work, we detailed the immediate energy and $L$-shell dependent response of the outer radiation belt to ICME-driven sheath regions. Our comprehensive statistical analysis showed the following:

1. The inner magnetospheric wave activity is enhanced during sheaths, including those sheaths that do not cause a notable geomagnetic disturbance. Similarly, non-geoeffective sheaths can also cause a significant response in the outer belt electron fluxes. This highlights the importance of considering also events with weak geomagnetic impact in studies of the outer radiation belt electron fluxes.

2. Electron flux enhancements occur predominantly in the heart of the outer belt at source and seed energies, while the dominant response of the core and ultrarelativistic population is depletion at high $L$-shells. Also the higher energy seed population is depleted at the highest sampled radial distances. These distinct results were specifically revealed by investigating, for the first time, the immediate, short timescale electron flux response.

Future work will make use of the phase space density analysis method (e.g., Green and Kivelson, 2001, 2004; Chen et al., 2005, 2007; Turner et al., 2012; Shprits et al., 2017), which excludes the effects of adiabatic processes, to study sheath response in more detail. With this method, the dominant acceleration, transport and loss processes in the outer radiation belt during sheath regions can be better identified. With the decommissioning of the Van Allen Probes, future missions surveying the radiation belt environment through various radial distances with high energy resolution are needed for the continuous study of the near-Earth space and its response to solar wind driving. In addition to large-scale missions such as the Van Allen Probes, radiation belt missions can be realised even with cost-effective nanosatellites (e.g., Palmroth et al., 2019).

*Data availability.* All RBSP-ECT data are publicly available at the website http://www.rbsp-ect.lanl.gov/, and all RBSP-EMFISIS data are publicly available at the website https://emfisis.physics.uiowa.edu. GOES data was obtained from the website https://www.ngdc.noaa.gov/stp/satellite/goes/index.html, and Wind and OMNI data were obtained from CDAWeb, https://cdaweb.gsfc.nasa.gov/index.html.

*Author contributions.* MK carried out the data analysis, prepared the plots and interpreted the results under the supervision of EK and AO. AJ helped in the interpretation of the results. MK prepared the manuscript with contributions from all authors.

*Competing interests.* The authors declare that they have no conflict of interest.

*Acknowledgements.* The results presented here have been achieved under the framework of the Finnish Centre of Excellence in Research

5     of Sustainable Space (FORESAIL; Academy of Finland grant number 312390), which we gratefully acknowledge. The work of LT is supported by the Academy of Finland (grant number 322544). Processing and analysis of the ECT, MagEIS and REPT data was supported by Energetic Particle, Composition, and Thermal Plasma (RBSP-ECT) investigation funded under NASA's Prime contract no. NAS5-01072. We acknowledge H. Spence and G. Reeves for the ECT data, B. Blake for the MagEIS data and D. Baker for the REPT data. We are also thankful to the Van Allen Probes, GOES, Wind and OMNI teams for making their data publicly available.

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
