# Peer review of "Outer radiation belt and inner magnetospheric response to sheath regions of coronal mass ejections: A statistical analysis"

_Annales Geophysicae, 2019_

## Referee Comment (RC1) · Anonymous Referee #1 · 28 Nov 2019

I have read the manuscript "Outer radiation belt and inner magnetospheric response to sheath regions of coronal mass ejections: A statistical analysis". The authors preform a very detailed study of sheath regions and how they affect the electron population of the outer radiation belt along with various geospace phenomena (e.g. EM waves, geomagnetic response, etc.). They also adopt a new approach considering not only storm events but also weak geomagnetic disturbances which I think it's quite important in order to gain a clear picture of the radiation belt response. I have several minor comments which I have commented and highlighted in the attached pdf but I also have some significant concerns mostly about the superposed epoch analysis and the way it is applied in the study.

[Figure]

MAJOR COMMENTS:

1) In page 5 lines 10-12 the authors report: ".  Therefore, we resampled the sheath regions to match the mean sheath duration of 12.0 h (Kilpua et al., 2013; Hietala et al., 2014).  The resampled data was acquired with linear interpolation." The re-sampling method needs more clarification and also justification of the use of linear interpolation. What is the mean duration of the sheaths under consideration and their standard deviation?  If the larger and the shorter duration of the events is comparable to the 12h duration (e.g.  14 and 10h respectively) then the linear interpolation gives you pretty good results. If not how can we be sure about the validity of the results?

2) In page 5 lines 21-25 the authors report: .  "For the ULF waves, we calculated the wavelet spectra for each three magnetic field components measured by GOES-15 and summed them together to estimate the total wave power spectral density. We calculated the Pc5 wave power in the range from 2.5 to 10 min (2–7 mHz) and the EMIC wave power in the range from 0.2 to 10 s (0.1–5 Hz), which corresponds to the range of Pc1 and Pc2 pulsations as given by Jacobs 25 et al. (1964)." Why the authors didn't use RBSP to obtain Pc1 and Pc5 power?  This way they could have a more straightforward comparison with chorus and fluxes which are obtained in the heart of the outer belt. Furthermore, Georgiou et al. 2018 (see figure 4 in the paper) performed a detailed statistical study with the use of epoch analysis and showed that there is a quite different evolution of Pc5 power beyond and below the geosynchronous orbit. Finally, why the authors apply wavelet analysis in each magnetic field component and then sum them? If they just want to see the total wave power it is more appropriate to apply the wavelet analysis in the magnitude of the magnetic field.

3) In section 3.3 lines 20-24 the authors report:"It is immediately evident that for geo-effective sheaths, enhancement events are more common at all energies and L-shells, and the source and seed populations are practically always enhanced in the heart of the outer belt (L = 3.5–5).  However, deviating from the superposed epoch analysis results, > MeV electrons experience depletion more frequently in geoeffective events

throughout the outer belt. In non-geoeffective events depletion begins to dominate the core population response only at around L > 5." Kilpua et al 2015 showed that there are significant flux dropouts during the sheath regions they examined. Can this be due to the 4h cadence you have chosen (I strongly believe that only 4 points during the sheath are very few in order to do statistics). If by choosing a higher resolution cadence you still don't see a dropout you need to argue about that. Another cause of that may be the averaging at L-shells. As you are showing in figure 6 there is significant depletion at L>4.5 but no depletion at L<4. In that case maybe you need to reapply the epoch analysis in different L=bins (e.g. 3-4, 4-5, 5-6). Of course this should be applied in waves as well.

MINOR COMMENTS:

1) page 2 line 3: "...storm and substorm processes, and by changes...", delete "end"

2) page 2 line 5: suggested reference: "Daglis IA, Katsavrias C, Georgiou M. 2019 From solar sneezing to killer electrons: outer radiation belt response to solar eruptions.Phil.Trans.R.Soc.A377: 20180097. http://dx.doi.org/10.1098/rsta.2018.0097"

3) page 2 line 6: suggested reference: "D.N. Baker, S.G. Kanekal, X. Li, S.P. Monk, J. Goldstein, J.L. Burch, Nature 432, 878 (2004). doi: 10.1038/nature03116"

4) page 2 line 21: There is reference to the ultra-relativistic population at the results section so I think it should be mentioned here as well (even though the boundary between relativistic and ultra-relativistic population is not well defined).

5) page 2 line 23: I believe that Jaynes et al. 2015 mentions that only seed electrons are accelerated by chorus.

6) page 2 line 28: It's not clear what that sentence mean. Does that implies that the density modulations produced by ULF waves can reduce the minimum electron energy for cyclotron resonance with EMIC waves? If yes please give reference to "Zhang, X.-J., Mourenas, D., Artemyev, A. V., Angelopoulos, V., & Sauvaud, J.-

A. (2019). Precipitation of MeV and sub-MeV electrons due to combined effects of EMIC and ULF waves. Journal of Geophysical Research: Space Physics, 124. https://doi.org/10.1029/2019JA026566".

7) page 2 line 30: Add "Jaynes, A. N., et al. (2014), Evolution of relativistic outer belt electrons during an extended quiescent period, J. Geophys. Res. Space Physics, 119, 9558–9566, doi:10.1002/2014JA020125."

8) page 2 line 31-33: I would suggest to separate references in a group which studies the response of the outer belt to storm events generally (e.g. Murphy 2018) and a group which studies the response due to different drivers (e.g. Kilpua 2015) and modify this paragraph accordingly. I would also suggest to include to references which correspond to the importance of source and seed population on the radiation belt dynamics:

Katsavrias, C., Daglis, I. A., & Li, W. (2019). On the statistics of acceleration and loss of relativistic electrons in the outer radiation belt: A superposed epoch analysis. Journal of Geophysical Research: Space Physics, 124. https://doi.org/10.1029/2019JA026569

Bingham, S. T., Mouikis, C. G., Kistler, L. M., Paulson, K. W., Farrugia, C. J., Huang, C. L., et al. (2019). The storm time development of source electrons and chorus wave activity during CME‐ and CIR‐driven storms. Journal of Geophysical Research: Space Physics, 124, 6438–6452. https://doi.org/10.1029/2019JA026689

9) page 3 line 15-17: I think that this is an important novelty of this work and should be further highlighted. It is well known that even weak or "non-storm" events can produce significant variability in the outer radiation belt population and that the Dst index can often not account for the internal mechanisms that are responsible for this variability. See also:

Schiller, Q., X. Li, L. Blum, W. Tu, D. L. Turner, and J. B. Blake (2014), A nonstorm time enhancement of relativistic electrons in the outer radiation belt, Geophys. Res. Lett., 41, 7–12, doi:10.1002/2013GL058485.

Katsavrias, C., I. A. Daglis, D. L. Turner, I. Sandberg, C. Papadimitriou, M. Georgiou, and G. Balasis (2015), Nonstorm loss of relativistic electrons in the outer radiation belt, Geophys. Res. Lett., 42, 10,521–10,530, doi:10.1002/2015GL066773.

10) page 4 line 21-23: Please modify according to the introduction. 1.5 MeV electrons are not considered as seed population but as core or relativistic. At the same extent 1.8 to 6.3 MeV electrons are relativistic and ultra-relativistic. Also, please clarify if you are using the background corrected fluxes from MagEIS.

11) page 5 line 17: Jaynes et al. 2014, among others, have shown that the effect of plasmaspheric hiss is significant at high energy electrons inside the plasmasphere and more important it is very slow (electron lifetimes down to 2.7 days at L=4.5). Is the study of such waves really necessary since you are studying sheaths which last for 12 hours?

12) page 5 line 28-29: The 4 hours binning provides you with ONLY 4 POINTS during the sheath region. Is that statistically enough?

13) page 5 line 30-31: There is a significant variability of the MagEIS lower energy channels up to September 2013 as discussed in Boyd et al. 2019. Does such a variability affect your data? If not, please argue.

Boyd,A.J.,Reeves,G.D.,Spence, H. E., Funsten, H. O., Larsen,B. A., Skoug, R. M., et al. (2019).RBSP-ECT combined spin-averagedelectron flux data product.Jour-nal of Geophysical Research: SpacePhysics,124. https://doi.org/10.1029/2019JA026733

14) page 6 line 15-16: The post-event flux is the average of the 12 h or the max or something else?

15) page 6 line 18: Again, do you mean the maximum flux during the sheath or some kind of averaging such as in the pre-event flux?

16) page 7 line 15: delete "SYM-H"

[Figure]

17) page 7 line 18-19: I don't understand the meaning of this sentence. You are referring to typical undisturbed condition but then you are talking about enhancement.

18) page 7 line 27-28: The format of the last panel does not allow the reader to discriminate the wave power enhancements. I believe it would be best if you showed Pc1 and Pc5 wave power separately.

19) page 8 line 19-20: I don't think this is accurate. As shown by the median, the substorm activity is pretty much comparable. The difference lies on the lower quantile.

20) page 8 line 30-31: Once again, if you consider the median, I believe that AL shows similar behavior during the sheath and during the ejecta which consequently explaines the behavior of chorus activity.

21) page 9 line 5-6: " That is, the median response of 346 keV electrons is an enhancement, as well, by a factor of about 8." Please rephrase.

22) In page 12 the authors report: "Interestingly, a feature in the outer belt response is that the depletion progresses to lower energies when L increases. At L âĹij 4.5 depletion dominates only at > 2 MeV energies, while at L âĹij 6 it has reached down to seed energies at around 500 keV. Depletion is most likely at high energies and high L-shells". This strongly indicates magnetopause shadowing effect.

23) page 12 line 6: Mention again your definition of geo-effectiveness

24) page 16 line 7-9: Add "Claudepierre, S. G., S. R. Elkington, and M. Wiltberger (2008), Solar wind driving of magnetospheric ULF waves:Pulsations driven by velocity shear at the magnetopause,J. Geophys. Res.,113, A05218, doi:10.1029/2007JA012890" as well as discussion about the generation process of ULF.

25) page 18 line 3-5: This is not correct. High energy electrons can penetrate deep inside the inner edge of the belt even during relatively weak events. For example, the relatively weak storm of April-May 2017 produce enhancements up to 10 MeV at

L=3-3.5

see for reference: "Katsavrias, C., Sandberg, I., Li, W., Podladchikova, O., Daglis, I. A., Papadimitriou, C., et al. (2019). Highly relativistic electron flux enhancement during the weak geomagnetic storm of April–May 2017. Journal of Geophysical Research: Space Physics, 124. https://doi.org/10.1029/ 2019JA026743

and

Zhao, H., Baker, D. N., Li, X., Jaynes, A. N., & Kanekal, S. G. (2018). The acceleration of ultrarelativistic electrons during a small to moderate storm of 21 April 2017. Geophysical Research Letters, 45, 5818–5825. https://doi.org/10.1029/2018GL078582

26) page 18 line 10-13: I don't understand the meaning of this sentence. If depletions are more pronounced with increasing energy and L-shell you have a clear indication for outward diffusion combined with magnetopause shadowing. Of course other wave particle interactions can contribute but at different energies and pitch angles each.

27) page 19: I would recommend to briefly summarize your most important results in bullets.

---

## Referee Comment (RC2) · Anonymous Referee #2 · 30 Nov 2019

This paper, entitled "Outer radiation belt and inner magnetospheric response to sheath regions of coronal mass ejections: a statistical analysis", shows the immediate response of inner magnetospheric plasma waves and electron fluxes to the driving of sheath regions preceding interplanetary coronal mass ejections. Through a superposed epoch analysis, the study shows the enhancements in wave powers of ULF, EMIC, chorus, and hiss waves during the sheaths compared to those during the preceding solar wind in both geoeffective and non-geoeffective events; source and seed populations often exhibit flux enhancements in the outer belt, while core and ultrarelativistic populations most exhibit flux decreases at high L region; and non-geoeffective sheaths can cause significant changes in the outer belt electron fluxes as

well. This study enriches and advances the results of previous studies on the effects of ICME/sheath on the inner magnetosphere by more strictly focusing on the sheath region, and the results shed light on the important effects of the sheath to the inner magnetosphere dynamics. This manuscript is overall well-written. However, there are still some concerns regarding the analysis method and interpretation of the results that I would like the authors to consider and address.

1. In the introduction, it is stated that "Our study includes sheaths that caused only a weak geomagnetic storm (-30 nT > SYM-H min > -50 nT) or no geomagnetic storm at all (SYM-H > -30 nT)" (line 27-28 on page 3). However, from Figure 4, it seems like the sheaths in some events did trigger stronger geomagnetic storms with SYM-H < -50 nT. Please check whether this is an inaccurate statement or Figure 4 needs to be corrected.

2. In this study, data from GOES-15 spacecraft were used for ULF and EMIC wave activity. However, the major results from this study focus on the dynamics of the inner magnetosphere at L<6. Since the wave distributions are L-dependent and localized, why not include measurements also from Van Allen Probes and other GOES satellites to enhance the spatial coverage?

3. On the other hand, the chorus and hiss wave activities were measured by the Van Allen Probes. Since chorus wave distribution is MLT-dependent, sampling may cause some bias in the statistical analysis. A plot of satellites' orbits in the statistical analysis will be helpful to reveal more detailed information.

4. The results of this study mainly focus on the electron flux variations. However, it is hard to isolate the effects of external drivers (i.e., sheath/ejecta) from adiabatic effects due to pure magnetic field configuration changes by only looking at electron fluxes. More discussions regarding the potential effects of adiabatic variations should be further discussed in the manuscript for both the event study and the statistical analysis.

5. Line 6-8 on page 17, "Since the sheaths cause enhancements in source electrons

but mostly depletion of more energetic seed electrons and the core population, they cannot, statistically, produce the so-called killer electrons (> 1–2 MeV), at least not under the studied timescales": From Figures 7 and 8, and also from the discussion earlier in this page, it seems like the seed electron fluxes enhanced in at least half of the cases. Especially during geoeffective events, despite the enhancements of both seed electron fluxes and chorus wave activity during sheaths, core population fluxes did not show enhancements. This indicates that the wave-particle interaction between chorus waves and seed populations may need a longer time to accelerate those electrons to MeV energies.

Minor issues:

1. It may be helpful to include some discussion on the results of previous studies on the ultrarelativistic electrons in the introduction. Only a small portion of geomagnetic storms in the Van Allen Probes era caused flux enhancements of ultrarelativistic electrons (e.g., Zhao et al., 2019), which may explain the results in this study that ultrarelativistic electrons often have little response to the sheaths.

Reference: Zhao, H., Baker, D. N., Li, X., Jaynes, A. N., & Kanekal, S. G. (2019). The effects of geomagnetic storms and solar wind conditions on the ultrarelativistic electron flux enhancements. Journal of Geophysical Research: Space Physics, 124. https://doi.org/10.1029/2018JA026257.

2. Line 3 on page 3, "The most important drivers of geomagnetic activity are interplanetary coronal mass ejections..." -> One of the most important drivers of...

3. About the solar wind data used in this study: have these observations been propagated to the bow shock nose?

4. Line 11-12 on page 5, "The resampled data was acquired with linear interpolation": Were electron fluxes also derived from linear interpolation? It makes more sense to linearly interpolate the logarithm of electron fluxes.

5. Section 3.2 and Figure 4: Some descriptions and discussions are needed for Figure 4(l) hiss waves. Also, it is confusing whether the mean or median of electron fluxes was used in this figure. From the text, it seems like the median of fluxes was used here, but the figure caption says the mean electron fluxes.

6. Line 8-9 on page 19, "...Reeves et al. (2013) showed that local acceleration, i.e., energization via wave-particle interactions, dominate in the heart of the outer belt" -> ... during an intense geomagnetic storm of October 2012.

---

## Author Comment (AC1) · 25 Jan 2020

**Referee #1**

I have read the manuscript "Outer radiation belt and inner magnetospheric response to sheath regions of coronal mass ejections: A statistical analysis". The authors preform a very detailed study of sheath regions and how they affect the electron population of the outer radiation belt along with various geospace phenomena (e.g. EM waves, geomagnetic response, etc.). They also adopt a new approach considering not only storm events but also weak geomagnetic disturbances which I think it's quite important in order to gain a clear picture of the radiation belt response. I have several minor comments which I have commented and highlighted in the attached pdf but I also have some significant concerns mostly about the superposed epoch analysis and the way it is applied in the study.

**We thank the referee for the constructive comments and suggestions that will improve the manuscript. Please find below our detailed responses.**

MAJOR COMMENTS:
1) In page 5 lines 10-12 the authors report: ". Therefore, we resampled the sheath regions to match the mean sheath duration of 12.0 h (Kilpua et al., 2013; Hietala et al., 2014). The resampled data was acquired with linear interpolation." The re-sampling method needs more clarification and also justification of the use of linear interpolation. What is the mean duration of the sheaths under consideration and their standard deviation? If the larger and the shorter duration of the events is comparable to the 12h duration (e.g. 14 and 10h respectively) then the linear interpolation gives you pretty good results. If not how can we be sure about the validity of the results?

**We have expanded the discussion on the superposed epoch analysis and mentioned the potential problems. We have emphasized that the sheaths were resampled to the mean sheath duration, which is 12.0 h. Additionally, we redid the superposed epoch analysis for sheaths whose duration was from 10 to 14 hours (in total 10 sheath events with the mean duration being 12.4 h). The results are similar to the results of the full set of 37 sheath events, which suggests that the resampling does not significantly affect the results.**

2) In page 5 lines 21-25 the authors report: . "For the ULF waves, we calculated the wavelet spectra for each three magnetic field components measured by GOES-15 and summed them together to estimate the total wave power spectral density. We calculated the Pc5 wave power in the range from 2.5 to 10 min (2–7 mHz) and the EMIC wave power in the range from 0.2 to 10 s (0.1–5 Hz), which corresponds to the range of Pc1 and Pc2 pulsations as given by Jacobs 25 et al. (1964)." Why the authors didn't use RBSP to obtain Pc1 and Pc5 power? This way they could have a more straightforward comparison with chorus and fluxes which are obtained in the heart of the outer belt. Furthermore, Georgiou et al. 2018 (see figure 4 in the paper) performed a detailed statistical study with the use of epoch analysis and showed that there is a quite different evolution of Pc5 power beyond and below the geosynchronous orbit. Finally, why the authors apply wavelet analysis in each magnetic field component and then sum them? If they just want to see the total wave power it is more appropriate to apply the wavelet analysis in the magnitude of the magnetic field.

**Calculating ULF wave power from RBSP data can be good for analysing local wave characteristics on shorter timescales, but the Van Allen Probes are not ideal for looking at long-term ULF wave statistics over the course of an event. The RBSP spacecraft move relatively fast through highly different plasma environments, observing vastly different**

**regions of the inner magnetosphere over the course of one half-orbit. GOES has the advantage of remaining at the same distance.**

**We have now discussed the effect of using ULF observations at geostationary orbit and included as references Georgiou et al. 2018 and Engebretson et al. 2018. We have also redone the wavelet analysis of ULF waves in the paper, using the magnitude of the magnetic field instead of the components. As a result, the wave power overall decreased but no significant changes were introduced in the profiles as the P-component always dominates the magnetic field magnitude.**

3) In section 3.3 lines 20-24 the authors report:"It is immediately evident that for geoeffective sheaths, enhancement events are more common at all energies and L-shells, and the source and seed populations are practically always enhanced in the heart of the outer belt (L = 3.5–5). However, deviating from the superposed epoch analysis results, > MeV electrons experience depletion more frequently in geoeffective events throughout the outer belt. In non-geoeffective events depletion begins to dominate the core population response only at around L > 5." Kilpua et al 2015 showed that there are significant flux dropouts during the sheath regions they examined. Can this be due to the 4h cadence you have chosen (I strongly believe that only 4 points during the sheath are very few in order to do statistics). If by choosing a higher resolution cadence you still don't see a dropout you need to argue about that. Another cause of that may be the averaging at L-shells. As you are showing in figure 6 there is significant depletion at L>4.5 but no depletion at L<4. In that case maybe you need to reapply the epoch analysis in different L=bins (e.g. 3-4, 4-5, 5-6). Of course this should be applied in waves as well.

**Kilpua et al. 2015 only studied >2 MeV electron fluxes at geostationary orbit and showed significant dropouts during sheath regions. Our results are in agreement with this study, as flux depletions dominate at MeV energies at the highest L-shells.**

**It is a very interesting question how the fluxes change within the sheath more precisely, but here we are mostly interested in how sheaths affect as a whole and what is the overall trend considering the pre- and post-sheath fluxes. This is also why we considered a wide L range (L = 3.5–5) for the superposed epoch analysis, and reserved the more detailed spatial (0.1 L) and temporal (1h) resolution for the electron flux response analysis. We also tried to use higher time resolution for superposed epoch analysis and the results are practically similar. We have now discussed this in the paper.**

MINOR COMMENTS:
1) page 2 line 3: "...storm and substorm processes, and by changes...", delete "end"

**Sentence was modified**

2) page 2 line 5: suggested reference: "Daglis IA, Katsavrias C, Georgiou M. 2019 From solar sneezing to killer electrons: outer radiation belt response to solar eruptions. Phil.Trans.R.Soc.A377: 20180097. http://dx.doi.org/10.1098/rsta.2018.0097"

**We have added this paper as a reference**

3) page 2 line 6: suggested reference: "D.N. Baker, S.G. Kanekal, X. Li, S.P. Monk, J. Goldstein, J.L. Burch, Nature 432, 878 (2004). doi: 10.1038/nature03116"

**We have added this reference**

4) page 2 line 21: There is reference to the ultra-relativistic population at the results section so I think it should be mentioned here as well (even though the boundary between relativistic and ultra-relativistic population is not well defined).

**Thank you for pointing this out. The ultrarelativistic population is now mentioned in the Introduction.**

5) page 2 line 23: I believe that Jaynes et al. 2015 mentions that only seed electrons are accelerated by chorus.

**This is correct. We have modified the sentence accordingly.**

6) page 2 line 28: It's not clear what that sentence mean. Does that implies that the density modulations produced by ULF waves can reduce the minimum electron energy for cyclotron resonance with EMIC waves? If yes please give reference to "Zhang, X.-J., Mourenas, D., Artemyev, A. V., Angelopoulos, V., & Sauvaud, J.-A. (2019). Precipitation of MeV and sub-MeV electrons due to combined effects of EMIC and ULF waves. Journal of Geophysical Research: Space Physics, 124. https://doi.org/10.1029/2019JA026566".

**Brito et al. 2012 suggest that ULF Pc4–Pc5 waves modulate the electron precipitation by lowering the mirror points. We have extended the discussion slightly to make the meaning clearer and added Zhang et al. 2019 as a reference.**

7) page 2 line 30: Add "Jaynes, A. N., et al. (2014), Evolution of relativistic outer belt electrons during an extended quiescent period, J. Geophys. Res. Space Physics, 119, 9558–9566, doi:10.1002/2014JA020125."

**Added**

8) page 2 line 31-33: I would suggest to separate references in a group which studies the response of the outer belt to storm events generally (e.g. Murphy 2018) and a group which studies the response due to different drivers (e.g. Kilpua 2015) and modify this paragraph accordingly. I would also suggest to include to references which correspond to the importance of source and seed population on the radiation belt dynamics:

Katsavrias, C., Daglis, I. A., & Li, W. (2019). On the statistics of acceleration and loss of relativistic electrons in the outer radiation belt: A superposed epoch analysis. Journal of Geophysical Research: Space Physics, 124. https://doi.org/10.1029/2019JA026569

Bingham, S. T., Mouikis, C. G., Kistler, L. M., Paulson, K. W., Farrugia, C. J., Huang, C. L., et al. (2019). The storm time development of source electrons and chorus wave activityduringCMEâA˘R˘andCIRâA˘R˘drivenstorms.JournalofGeophysicalResearch: Space Physics, 124, 6438–6452. https://doi.org/10.1029/2019JA026689

**Thank you for this good suggestion. We have divided the references into two groups and added a discussion of the importance of the source and seed populations. The two papers were added as references.**

9) page 3 line 15-17: I think that this is an important novelty of this work and should be further highlighted. It is well known that even weak or "non-storm" events can produce significant variability in the outer radiation belt population and that the Dst index can often not account for the internal mechanisms that are responsible for this variability. See also:

Schiller, Q., X. Li, L. Blum, W. Tu, D. L. Turner, and J. B. Blake (2014), A nonstorm time enhancement of relativistic electrons in the outer radiation belt, Geophys. Res. Lett., 41, 7–12, doi:10.1002/2013GL058485.

Katsavrias, C., I. A. Daglis, D. L. Turner, I. Sandberg, C. Papadimitriou, M. Georgiou, and G. Balasis (2015), Nonstorm loss of relativistic electrons in the outer radiation belt, Geophys. Res. Lett., 42, 10,521–10,530, doi:10.1002/2015GL066773.

**We have further highlighted the inclusion of weak and non-storm events in our study and added the suggested two papers as references.**

10) page 4 line 21-23: Please modify according to the introduction. 1.5 MeV electrons are not considered as seed population but as core or relativistic. At the same extent 1.8 to 6.3 MeV electrons are relativistic and ultra-relativistic. Also, please clarify if you are using the background corrected fluxes from MagEIS.

**Thank you for pointing this out. We have modified the sentence. Thank you also very much for pointing out the background correction. We were indeed using the uncorrected MagEIS electron flux measurements. We will redo the MagEIS electron flux analysis using the background corrected measurements.**

11) page 5 line 17: Jaynes et al. 2014, among others, have shown that the effect of plasmaspheric hiss is significant at high energy electrons inside the plasmasphere and more important it is very slow (electron lifetimes down to 2.7 days at L=4.5). Is the study of such waves really necessary since you are studying sheaths which last for 12 hours?

**We have included the study of hiss waves among with other wave modes for completeness. We have now mentioned this in the paper with a reference to Jaynes et al. 2014.**

12) page 5 line 28-29: The 4 hours binning provides you with ONLY 4 POINTS during the sheath region. Is that statistically enough?

**See our response above for major comment 3**

13) page 5 line 30-31: There is a significant variability of the MagEIS lower energy channels up to September 2013 as discussed in Boyd et al. 2019. Does such a variability affect your data? If not, please argue.

Boyd,A.J.,Reeves,G.D.,Spence, H. E., Funsten, H. O., Larsen,B. A., Skoug, R. M., et al. (2019).RBSP-ECT combined spin-averaged electron flux data product.Journal of Geophysical Research: SpacePhysics,124. https://doi.org/10.1029/2019JA026733

**We have in our study 13 events before September 2013. We have now mentioned this possibility in the paper and included Boyd et al. 2019 as a reference.**

14) page 6 line 15-16: The post-event flux is the average of the 12 h or the max or something else?

**The post-event flux is a 6-hour average after the sheath, similarly to the pre-event flux which is a 6-hour average before the sheath. We modified the sentence to make the definitions of the pre- and post-event fluxes clearer.**

15) page 6 line 18: Again, do you mean the maximum flux during the sheath or some kind of averaging such as in the pre-event flux?

**See above**

16) page 7 line 15: delete "SYM-H"

**Deleted**

17) page 7 line 18-19: I don't understand the meaning of this sentence. You are referring to typical undisturbed condition but then you are talking about enhancement.

**Changed "electrons being enhanced" to "electron fluxes being higher"**

18) page 7 line 27-28: The format of the last panel does not allow the reader to discriminate the wave power enhancements. I believe it would be best if you showed Pc1 and Pc5 wave power separately.

**We have added a panel to Figure 3 showing the wave power of ULF Pc5 and EMIC waves during the event.**

19) page 8 line 19-20: I don't think this is accurate. As shown by the median, the substorm activity is pretty much comparable. The difference lies on the lower quantile.

**We have added that the substorm activity shown by AL index is weak during both the sheath and ejecta.**

20) page 8 line 30-31: Once again, if you consider the median, I believe that AL shows similar behavior during the sheath and during the ejecta which consequently explaines the behavior of chorus activity.

**We agree with the referee and have changed the discussion accordingly.**

21) page 9 line 5-6: " That is, the median response of 346 keV electrons is an enhancement, as well, by a factor of about 8." Please rephrase.

**Rephrased to "The 346 keV electron median flux increases by a factor of about 8."**

22) In page 12 the authors report: "Interestingly, a feature in the outer belt response is that the depletion progresses to lower energies when L increases. At L âˊLij 4.5 depletion dominates only at > 2 MeV energies, while at L âˊLij 6 it has reached down to seed energies at around 500 keV. Depletion is most likely at high energies and high L-shells". This strongly indicates magnetopause shadowing effect.

**We have suggested in the discussion that magnetopause shadowing is an explanation for the losses at high L-shells.**

23) page 12 line 6: Mention again your definition of geo-effectiveness

**Added**

24) page 16 line 7-9: Add "Claudepierre, S. G., S. R. Elkington, and M. Wiltberger (2008), Solar wind driving of magnetospheric ULF waves:Pulsations driven by velocity shear at the magnetopause,J. Geophys. Res.,113, A05218, doi:10.1029/2007JA012890" as well as discussion about the generation process of ULF.

**We will add this reference and discuss shortly in the Introduction how ULF waves are typically generated.**

25) page 18 line 3-5: This is not correct. High energy electrons can penetrate deep inside the inner edge of the belt even during relatively weak events. For example, the relatively weak storm of April-May 2017 produce enhancements up to 10 MeV at L=3-3.5

see for reference: "Katsavrias, C., Sandberg, I., Li, W., Podladchikova, O., Daglis, I. A., Papadimitriou, C., et al. (2019). Highly relativistic electron flux enhancement during the weak geomagnetic storm of April–May 2017. Journal of Geophysical Research: Space Physics, 124. https://doi.org/10.1029/2019JA026743

and

Zhao, H., Baker, D. N., Li, X., Jaynes, A. N., & Kanekal, S. G. (2018). The acceleration of ultrarelativistic electrons during a small to moderate storm of 21 April 2017. Geophysical Research Letters, 45, 5818–5825. https://doi.org/10.1029/2018GL078582

**We have now mentioned that high-energy electrons can penetrate to lower L-shells also during weak storms and added the two papers as references.**

26) page 18 line 10-13: I don't understand the meaning of this sentence. If depletions are more pronounced with increasing energy and L-shell you have a clear indication for outward diffusion combined with magnetopause shadowing. Of course other wave particle interactions can contribute but at different energies and pitch angles each.

**We agree with the referee that outward diffusion could play a role in concert with magnetopause shadowing at higher L-shells. We will discuss this in more detail in the revised manuscript.**

27) page 19: I would recommend to briefly summarize your most important results in bullets.

**We have now written the summary in bullet points.**

---

## Author Comment (AC2) · 25 Jan 2020

**Referee #2**

This paper, entitled "Outer radiation belt and inner magnetospheric response to sheath regions of coronal mass ejections: a statistical analysis", shows the immediate response of inner magnetospheric plasma waves and electron fluxes to the driving of sheath regions preceding interplanetary coronal mass ejections. Through a superposed epoch analysis, the study shows the enhancements in wave powers of ULF, EMIC, chorus, and hiss waves during the sheaths compared to those during the preceding solar wind in both geoeffective and non-geoeffective events; source and seed populations often exhibit flux enhancements in the outer belt, while core and ultrarelativistic populations most exhibit flux decreases at high L region; and non- geoeffective sheaths can cause significant changes in the outer belt electron fluxes as well. This study enriches and advances the results of previous studies on the effects of ICME/sheath on the inner magnetosphere by more strictly focusing on the sheath region, and the results shed light on the important effects of the sheath to the inner magnetosphere dynamics. This manuscript is overall well-written. However, there are still some concerns regarding the analysis method and interpretation of the results that I would like the authors to consider and address.

**We thank the referee for the constructive comments and corrections that will improve the manuscript. Please find below our detailed responses.**

1. In the introduction, it is stated that "Our study includes sheaths that caused only a weak geomagnetic storm (-30 nT > SYM-H min > -50 nT) or no geomagnetic storm at all (SYM-H > -30 nT)" (line 27-28 on page 3). However, from Figure 4, it seems like the sheaths in some events did trigger stronger geomagnetic storms with SYM-H < -50 nT. Please check whether this is an inaccurate statement or Figure 4 needs to be corrected.

**The study includes events that caused weak or no storms, as well as events causing stronger geomagnetic activity. We have clarified the statement on page 3.**

2. In this study, data from GOES-15 spacecraft were used for ULF and EMIC wave activity. However, the major results from this study focus on the dynamics of the inner magnetosphere at L<6. Since the wave distributions are L-dependent and localized, why not include measurements also from Van Allen Probes and other GOES satellites to enhance the spatial coverage?

**Calculating ULF wave power from RBSP data can be good for analysing local wave characteristics on shorter timescales, but the Van Allen Probes are not ideal for looking at long-term ULF wave statistics over the course of an event. The RBSP spacecraft move relatively fast through highly different plasma environments, observing vastly different regions of the inner magnetosphere over the course of one half-orbit. GOES has the advantage of remaining at the same distance. We will look into the possibility of including ULF observations from more than one GOES satellite.**

**We have now discussed the effect of using ULF observations from a geostationary GOES satellite and referenced to Georgiou et al. 2018 and Engebretson et al. 2018.**

3. On the other hand, the chorus and hiss wave activities were measured by the Van Allen Probes. Since chorus wave distribution is MLT-dependent, sampling may cause some bias in the statistical analysis. A plot of satellites' orbits in the statistical analysis will be helpful to reveal more detailed information.

**We have considered the referee's suggestion, but feel that there is no feasible way to provide orbit information for this type of statistical analysis. In the statistical superposed epoch analysis, the median wave power is calculated from the data of 37 events at each time step, which most likely averages out MLT dependence.**

4. The results of this study mainly focus on the electron flux variations. However, it is hard to isolate the effects of external drivers (i.e., sheath/ejecta) from adiabatic effects due to pure magnetic field configuration changes by only looking at electron fluxes. More discussions regarding the potential effects of adiabatic variations should be further discussed in the manuscript for both the event study and the statistical analysis.

**The referee makes an excellent point, but we would like to reserve electron phase space density analysis for a more detailed study in the future. The method has been contemplated on a general level in the Discussion.**

5. Line 6-8 on page 17, "Since the sheaths cause enhancements in source electrons but mostly depletion of more energetic seed electrons and the core population, they cannot, statistically, produce the so-called killer electrons (> 1–2 MeV), at least not under the studied timescales": From Figures 7 and 8, and also from the discussion earlier in this page, it seems like the seed electron fluxes enhanced in at least half of the cases. Especially during geoeffective events, despite the enhancements of both seed electron fluxes and chorus wave activity during sheaths, core population fluxes did not show enhancements. This indicates that the wave-particle interaction between chorus waves and seed populations may need a longer time to accelerate those electrons to MeV energies.

**This was indeed written unclearly. We have reformulated the sentence to reflect that sheaths cause enhancements of the seed population but that depletion dominates at the highest > 500 keV seed energies.**

Minor issues:
1. It may be helpful to include some discussion on the results of previous studies on the ultrarelativistic electrons in the introduction. Only a small portion of geomagnetic storms in the Van Allen Probes era caused flux enhancements of ultrarelativistic electrons (e.g., Zhao et al., 2019), which may explain the results in this study that ultrarelativistic electrons often have little response to the sheaths.
Reference: Zhao, H., Baker, D. N., Li, X., Jaynes, A. N., & Kanekal, S. G. (2019). The effects of geomagnetic storms and solar wind conditions on the ultrarelativistic electron flux enhancements. Journal of Geophysical Research: Space Physics, 124. https://doi.org/10.1029/2018JA026257.

**We have now discussed the results of Zhao et al. 2019 in the Discussion.**

2. Line 3 on page 3, "The most important drivers of geomagnetic activity are interplanetary coronal mass ejections. . ." -> One of the most important drivers of. . .

**Corrected**

3. About the solar wind data used in this study: have these observations been propagated to the bow shock nose?

**Yes, thank you for noticing this. We have added this information to Section 2.1.**

4. Line 11-12 on page 5, "The resampled data was acquired with linear interpolation": Were electron fluxes also derived from linear interpolation? It makes more sense to linearly interpolate the logarithm of electron fluxes.

**Thank you for this suggestion. We have now interpolated the logarithm of electron fluxes as well as the logarithm of wave power.**

5. Section 3.2 and Figure 4: Some descriptions and discussions are needed for Figure 4(l) hiss waves. Also, it is confusing whether the mean or median of electron fluxes was used in this figure. From the text, it seems like the median of fluxes was used here, but the figure caption says the mean electron fluxes.

**We have added a discussion of hiss waves in Section 3.2. Figure 4 shows median electron fluxes. Thank you for noticing the typo, we have now corrected the caption of Figure 4.**

6. Line 8-9 on page 19, ". . .Reeves et al. (2013) showed that local acceleration, i.e., energization via wave-particle interactions, dominate in the heart of the outer belt" -> . . . during an intense geomagnetic storm of October 2012.

**Added**

---

## Author Response (AR2)

**Manuscript angeo-2019-150: "Outer radiation belt and inner magnetospheric response to sheath regions of coronal mass ejections: A statistical analysis" by Milla M. H. Kalliokoski et al.**

We thank both referees for the careful consideration of our manuscript. We have provided our point-by-point responses to the comments by Referee #1 below, where we have indicated the referee comments in plain text and our responses **in bold**. The corresponding modifications appear **in bold** in the revised manuscript.

**Response to Referee #1**

1) Page 1, line 4: "...in the outer belt to driving by...". Change to "...in the outer belt driven by..."

**Corrected**

2) Page 6, line 29: " In previous studies, events have typically been divided with the threshold of −50 nT, or only these moderate or larger storms are considered." Please rephrase.

**Rephrased to "In previous studies, typically only moderate or larger storms with Dst or SYM-H less than -50 nT have been considered" (p. 6, l. 29–30).**

3) Figure 4, 7 and 8: Please re-scale y-axis in order to include all values (for example AL in figure 7).

**We have re-scaled the y-axes of Figures 4, 7 and 8.**

4) Page 18, line 1-3: "We also note that in the solar wind, as opposed to the magnetosphere as investigated here, sheaths have in general a clearly higher level of ULF Pc5 wave power than the ejecta and the preceding solar wind (Kilpua et al., 2013; Hietala et al., 2014)". This sentence is a little bit confusing. Please rephrase.

**We have rephrased the sentence as follows (p. 18, l. 1–4):**

***"In this study, we found enhanced ULF wave activity during sheaths as observed inside the magnetosphere, and we note that Kilpua et al. (2013) and Hietala et al. (2014) have observed in general a clearly higher level of ULF Pc5 wave power during sheaths than during ejecta and the preceding solar wind also outside the magnetosphere."***

5) Page 18, line 10: "we detailed more precisely the more immediate sheath response...". Please rephrase.

**Rephrased to "we examined more precisely the immediate, few hours timescale sheath response" (p. 18, l. 11).**

6) Page 19, line 24: "electron fluxes at seed energies" change to "seed electron fluxes"

**Changed**

7) Page 19, line 26: "L-shells dependence" change to "L-shell dependence"

**Corrected**

8) Page 20, line 5-10: You cannot really compare PSD with flux studies. Also I don't see the point in comparing Reeves et al. 2013 since they studied an intense storm while you have studied a sample of moderate or week disturbances. Please rephrase the whole paragraph.

**We have modified the paragraph to emphasize that this is not a rigorous comparison between our study and phase space density studies and mentioned that we also study geomagnetically weak events. The modified paragraph is as follows (p. 20, l. 6–12):**

*"The results in this paper agree qualitatively with the general conclusions of phase space density studies. However, we note that these studies are not quantitatively comparable with ours since we examined electron fluxes and considered also non-geoeffective events. During an intense geomagnetic storm, Reeves et al. (2013) showed using phase space density analysis 
[revised manuscript text omitted]